



# Global distribution of $CO_2$ in the Upper-Troposphere and Stratosphere

**M. Diallo[1,2], B. Legras[1], E. Ray[3,4], A. Engel[5], and J. A. Añel[2,6]**

[1]Laboratoire de Météorologie Dynamique, UMR8539, IPSL, UPMC/ENS/CNRS/Ecole Polytechnique, Paris, France
[2]EPhysLab, Facultade de ciencias, Universidade de Vigo, Ourense, Spain
[3]Chemical Sciences Division, Earth Systems Research Laboratory, NOAA, Boulder, Colorado, USA
[4]Cooperative Institute for Research in Environmental Sciences,University of Colorado, Boulder, Colorado, USA
[5]Institute for Atmospheric and Environmental Sciences, Goethe University Frankfurt, Frankfurt am Main, Germany
[6]Smith School of Enterprise and the Environment, University of Oxford, Oxford, UK

Correspondence to: M. Diallo (mdiallo@lmd.ens.fr)



## Abstract

In this study, we aim to reconstruct a relevant and new database of monthly zonal mean distribution of *carbon dioxide* ($CO_2$) at global scale extending from the *upper-troposphere* (UT) to *stratosphere* (S). This product can be used for model and satellite validation in the UT/S, as a prior for inversion modelling and mainly to analyse a plausible feature of the stratospheric-tropospheric exchange as well as the stratospheric circulation and its variability. To do so, we investigate the ability of a Lagrangian trajectory model guided by ERA-Interim reanalysis to construct the $CO_2$ abundance in the UT/S. From 10 year backward trajectories and tropospheric observations of $CO_2$, we reconstruct upper-tropospheric and stratospheric $CO_2$ over the period 2000–2010. The inter-comparisons of the reconstructed $CO_2$ with mid-latitude vertical profiles measured by balloon samples as well as quasi-horizontal air samples from ER-2 aircraft during SOLVE and CONTRAIL campaigns exhibit a remarkable agreement. That demonstrates the potential of Lagrangian model to reconstruct $CO_2$ in the UT/S. The zonal mean distribution exhibits relatively large $CO_2$ in the tropical stratosphere due to the seasonal variation of the tropical upwelling of Brewer-Dobson circulation. During winter and spring, the *tropical pipe* is relatively isolated but is less confined during summer and autumn so that high $CO_2$ values are more readily transported out of the tropics to the mid- and high latitude stratosphere. The shape of the vertical profiles suggests that relatively high $CO_2$ above 20 km altitude mainly enter the stratosphere through tropical upwelling. $CO_2$ mixing ratio is relatively low in the polar and tropical regions above 25 km. On average the $CO_2$ mixing ratio decreases with altitude by 6-8 ppmv from the UT to stratosphere (e.g. up to 35 km) and is nearly constant with altitude.

## 1 Introduction

Over the last twenty years, climate change has made the stratosphere a focus of several debates among the scientific community. The global stratospheric meridional circulation, also called *Brewer-Dobson circulation* (BDC), has been recognized as a major component of the climate system, which affects radiative forcing (Lacis et al., 1990; Forster and Shine, 1997; Forster





et al., 2007) and atmospheric circulation (Andrews et al., 1987; Holton et al., 1995; Salby and Callaghan, 2005, 2006). The increase of the greenhouse gases, in particular *carbon dioxide* ($CO_2$) concentration, increases the tropical upwelling mass flux (Butchart et al., 2010; Garny et al., 2011; Abalos et al., 2015) and therefore changes the BDC.

$CO_2$ is a good tracer of the atmospheric dynamics and transport because of its long lifetime in the atmosphere where it has essentially no sources or sinks and as it is chemically inert in the free troposphere. The only stratospheric source of $CO_2$ is a small contribution from methane ($CH_4$) oxidation that is on the order of $1\,\mathrm{ppmv}$ (i.e. parts per million in volume) (Boucher et al., 2009). $CO_2$ is regularly exchanged between these four reservoirs: among the biosphere (photosynthesis and respiration), lithosphere (soil and fossil pool), hydrosphere (surface and deep ocean), and atmosphere with a time residency much longer in the ocean and soil than in the atmosphere. These exchanges are described as the carbon cycle. Clearly, the increase of the anthropogenic emissions, deforestation and biomass and fossil fuel burning have systematically increased the mean $CO_2$ growth rate and modified its season cycle during these last two decades (Tans and Keeling, 2015). With steady growth and seasonal variation, $CO_2$ contains a double time signature: (a) monotonically increasing and (b) periodic tracer signal that represents stringent tests of stratospheric transport and *Stratosphere-Troposphere Exchange* (STE) in models (Waugh and Hall, 2002; Bönisch et al., 2008, 2009).

Despite its potential to increase global warming by cooling the stratosphere and warming the troposphere, until recently, our knowledge of stratospheric $CO_2$ abundances and variability was sparse. In recent years several *in situ* aircraft and balloon campaigns were held to measure a number of chemical tracers including $CO_2$. The *in situ* campaigns included SPURT aircraft measurements in the *upper-troposphere and lower stratosphere* (UTLS) (Engel et al., 2006; Gurk et al., 2008), CONTRAIL (Sawa et al., 2008) and CARIBIC (Schuck et al., 2009; Sprung and Zahn, 2010). Although sporadic in time and space coverage, these *in situ* measurements have been used to analyse the BDC changes (Andrews et al., 2001a; Engel et al., 2009; Ray et al., 2014), to validate Chemistry-Transport Models (CTMs) (Strahan et al., 2007; Waugh, 2009) and to diagnose STE (Strahan et al., 1998; Bönisch et al., 2009, 2011). The *in situ* campaigns, such as SPURT, were used in previous studies to diagnose the eddy diffusivity in



the *lowermost stratosphere* (LMS) using a 2D-advection-diffusion model (Hegglin et al., 2005) and also to quantify different transport pathways contributing to the composition of the LMS over Europe (Hoor et al., 2005, 2010). These studies helped to evolve our understanding on a finite chemical transition layer in the extra-tropical tropopause layer and identified potential gaps in our understanding. The stratospheric overworld circulation changes that affect the extra-tropical UTLS was recently assessed from balloon long-term measurements of stratospheric age tracers, such as $SF_6$ and $CO_2$. The stratospheric mean *age of air* (AoA), that is defined as the time residence of air parcels in the stratosphere (Li and Waugh, 1999; Waugh and Hall, 2002; Butchart et al., 2010), was calculated by Ray et al. (2014) more consistently from the *in situ* profile measurements of trace gases with an idealized model to identify the natural variability in the BDC and its significant linear trends. This study exhibited the importance of reconstructed *in situ* measurements to help validate the stratospheric portion of global CTMs and Chemistry Climate Models (CCMs).

In addition to the *in situ* observations that are high resolution and very localised, a large spatial and temporal coverage of $CO_2$ is obtained from space by satellite observations from vertical nadir sounders TOVS (Chedin et al., 2002a,b, 2003b), AIRS (Chedin et al., 2003a), SCIAMACHY (Bowman et al., 2000), IASI (Chedin et al., 2003a), GOSAT (Hammerling et al., 2012; Liu et al., 2014) and recently OCO-2 (Frankenberg et al., 2015). Chedin et al. (2003a) showed that upper air $CO_2$ can be retrieved from observations of TOVS, whose main mission is to measure atmospheric temperature and moisture at global scale. Foucher et al. (2009, 2011) have obtained five years of monthly mean $CO_2$ vertical profiles by analysing the ACE-FTS data (Bernath et al., 2005). This space-borne instrument is passive because it uses the sunset and sunrise to measure the atmospheric chemistry species, such as isotopologues, in limb-view. The main isotopologue $^{12}C^{16}O_2$ passively measured in limb-view was used to retrieve the mean $CO_2$ in the 10–25 km altitude range. When the main isotopologue lines saturate at low altitude range from 5 to 15 km, the $^{18}OC^{16}O$ isotopologue was used (more detail about the inversion approach is found in Foucher et al. (2009)). The ACE-retrieved $CO_2$ has shown a good agreement after its qualitatively validation with *in situ* observations for 2004–2008 time period and at the 50 °N–60 °N latitude bins. The ACE-FTS retrieval approach limitation was due to the dif-



ficulty to have an accurate high sensitivity of $N_2$ continuum absorption, which allows to obtain a better tangent height retrieval. The observations from space-borne instruments are oftentimes hampered by the fact that these instruments measure an integrated column, which can not be used to better understand the $CO_2$ distribution mechanisms. The $CO_2$ column measurements, essentially weighted from the middle to upper-troposphere, are also weak and not easy to interpret globally. Because of these limitations in the observations, the CTMs and Lagrangian transport models combined with these observations are a useful and complementary framework to widely diagnose the BDC and to represent the global transport and distribution of long-lived species, such as $CO_2$, within the UT/S.

Previous studies using the two-dimensional CTM Caltech/JPL (Shia et al., 2006) and the three-dimensional CTM TM5 (Transport Model 5) (Bönisch et al., 2008) were unable to accurately represent the BDC. Bönisch et al. (2008) investigated the UTLS exchanges in a three-dimensional transport model using the observed $CO_2$ and $SF_6$ distributions and concluded that major problems occur in winter, where a too strong BDC leads to some overestimates of the $CO_2$, and in boreal summer, where a too weak vertical transport arise in the UT. During autumn, the models showed a persistent reverse gradient in the lower stratosphere due to an unrealistic BDC and during spring, the transport processes through the tropical tropopause are overestimated, which explained the high $CO_2$ values in the lower stratosphere. Furthermore, many three-dimensional models are too diffusive and/or have too strong mixing when crossing the tropopause that lead to an underestimation of the amplitude of the seasonal cycle in the column of $CO_2$ (Olsen and Randerson, 2004). Shia et al. (2006) suggested that the model's lack of realistic stratospheric influence on the $CO_2$ could explain part of this discrepancy. The persistence of the inverted $CO_2$ gradient noted in these models can result in an underestimation of the exchange of air masses from the stratosphere to the troposphere in mid-latitudes during fall. Models also fail to simulate a realistic $CO_2$ seasonal cycle and STE because also of short time period of simulation. Shia et al. (2006) concluded that at least three years are required for the surface $CO_2$ to be transported into upper-troposphere and LMS then moved to the temperate and polar latitudes. In order to eliminate diffusivity effect, Lagrangian or quasi-Lagrangian model, such as CLaMS and TRACZILLA, have been widely used to investigate the transport





problem. The combination of these Lagrangian models with *in situ* observations to reconstruct chemical tracer gas significantly contributed to evolve our understanding on quantifying mixing effect to STE across the extra-tropical tropopause (Hoor et al., 2004; Pan et al., 2006; James and Legras, 2009), on analysis of the filamentary structures in both long-lived and chemically active
5   species near the edge of the polar vortex (Konopka et al., 2003), on evaluating of the transport of carbon monoxide and long-lived trace species from the tropical troposphere to stratosphere (Pommrich et al., 2014) and finally on quantifying the processes that control UTLS composition (Riese et al., 2012).

    The small scale variability in the chemical tracer gas, particularly $CO_2$ here, its strong gradi-
10  ents across the tropopause and its scarcity suitable for validation purposes lead to a challenging task for its validation in the UTLS as well as for CTMs and CCMs to reproduce well its feature (Hegglin et al., 2008).

    In this paper our goal is to reconstruct a new relevant database of monthly zonal mean distribution of $CO_2$ at global scale extending from the upper-troposphere to stratosphere using
15  a Lagrangian model guided by ERA-Interim reanalysis. This product can be used to validate CTMs, CCMs, as a prior for inversion modelling and mainly to analyse a plausible feature of the stratospheric-tropospheric exchange as well as the stratospheric circulation and its variability.

    To do so, here, we reconstruct a global distribution of $CO_2$ calculated over the time period 2000-2010 from a Lagrangian transport model driven by the most recent reanalysed winds
20  and heating rates from the ERA-Interim reanalysis of the European Centre for Medium Range Weather Forecast (ECMWF). The global distribution of $CO_2$ and its transport in the UT/S are investigated using backward deterministic trajectories, which are integrated over 10 years in time. We describe the method and data used in this study in Sect. 2 and Sect. 3, respectively. The reconstruction techniques of the *in situ* balloon and aircraft observations are detailed in
25  Sect. 4. The reconstructed $CO_2$ are compared with observations in Sect. 5. The time series in the northern hemisphere over $10°$ of latitudes bins are presented in Sect. 5.3. The global monthly distribution of the zonal mean $CO_2$ and its variability are discussed in Sect. 6. Finally, section 7 provides further discussions and conclusions.



## 2   Backward trajectory method

Backward deterministic trajectories are calculated using the Lagrangian transport model (Legras et al., 2005), which is a modified version of FLEXPART (Stohl et al., 2005). TRACZILLA uses analysed winds to move particles in the horizontal direction and performs direct interpolation from data on hybrid levels. In the vertical direction, it uses either pressure coordinate and Lagrangian pressure tendencies, or potential temperature coordinate and heating rates. We denote the trajectories as *diabatic* following a convention established by Eluszkiewicz et al. (2000). At each level in the vertical, air particles are initialised over a longitude-latitude grid with $2\,^\circ$ resolution in latitude and an almost uniform spacing in longitude of $2\,^\circ/\cos(\phi)$, where $\phi$ is the latitude, generating $10\,255$ particles on each level from pole to pole. To avoid interpolation errors, the vertical levels of the initial grid are chosen to be the hybrid levels of the ECMWF model. In order to encompass the whole stratosphere at any latitude, the 30 levels from about $400\,\mathrm{hPa}$ (varying according to the surface pressure) to $2\,\mathrm{hPa}$ are selected. Trajectories starting below the tropospheric boundary condition, which is defined as the level at which we assign the free tropospheric $CO_2$ to air particles, are immediately stopped and therefore do not induce any computational cost. Particles starting in the stratosphere are integrated backward in time until they cross the tropopause, defined as the lower envelope of the surfaces $\theta=380\,\mathrm{K}$ and $|P| = 2 \times 10^{-6}\,\mathrm{K\,kg^{-1}\,m^2\,s^{-1}}$ where $P$ is the Ertel potential vorticity (see Sect. 4.1) and reach our tropospheric boundary condition. Ensembles of particles were launched at the end of every month over the period 2000–2010 and were integrated backwards for 10 years (Diallo et al., 2012).

## 3   Data

To reconstruct the global distribution of $CO_2$ from UT to the stratosphere, we need to assign the backward trajectories with the free tropospheric $CO_2$ field. This is achieved by using two different types of $CO_2$ data: ground stations (Worden et al., 2015) and CarbonTracker (Peters et al., 2007). We first focus on the free tropospheric $CO_2$ field.





## 3.1 Construction of the free tropospheric $CO_2$ field

Two different observation-based data sets are used to assign $CO_2$ to the backward trajectories in the Lagrangian model depending on the time period considered.

During the 1989–1999 time period, data from ground stations of the World Data Centre for Greenhouse Gases (WDCGG) are used to assign the integrated air particles. The WDCGG is an international data center participating in Global Atmosphere Watch. It provides extensive data from ground stations and aircraft measurements across the Earth that are non-homogeneously distributed. To model the $CO_2$ emissions, the monthly $CO_2$ data from ground stations (e.g. Mauna Loa, South Pole and others) located at different latitudes have been used to overcome the daily fluctuations of the $CO_2$ in the atmospheric boundary layer. The criterion to select the ground stations is that the location is assured to be high enough to neglect the daily variability due to the contribution of localized sources at ground level. The $CO_2$ data are averaged from pole to pole in latitude increments of $30°$ and over all longitudes to represent the global, free tropospheric $CO_2$ field. To better model the latitude dependence of the seasonal cycle and to overcome discontinuities, the averaged $CO_2$ data obtained are then interpolated linearly along the latitude. This is our baseline for injecting $CO_2$ values to constrain the backward trajectories from the model during the 1989–1999 time period.

During the remaining time period 2000–2010, we use the most recent $CO_2$ mole fractions released in a longitude $\times$ latitude grid ($3° \times 2°$) from the coupled CTM TM5 and the data assimilation system, CarbonTracker, to assign the backward trajectories. Similar to the 1989–1999 time period, we consider these $CO_2$ from CarbonTracker only at $5\,km$ level ($500\,hPa$), which is above the atmospheric boundary layer. The CarbonTracker system assimilates different types of $CO_2$ measurements, which come from ocean surface fluxes, atmospheric stations, biosphere activity monitoring, fire burning and fossil fuel burning emissions inventoried in many regions. These data are important to achieve a complete and realistic diagnostic of the atmospheric $CO_2$ and fluxes (CarbonTracker-2013B, www.esrl.noaa.gov/gmd/ccgg/carbontracker/). This last archive of $CO_2$ is more accurate than the previous because of the recent correction of TM5 in reproducing a realistic convective flux due to a fault in one of the vertical mixing





parametrisations of the model.

## 3.2 *In situ* aircraft and balloon measurements

The SAGE III Ozone Loss and Validation Experiment (SOLVE) was a measurement campaign designed to examine the processes controlling ozone levels at mid- to high latitudes. Measure-
ments were made in the Arctic high-latitude region in winter using the NASA DC-8 and ER-2 aircrafts, as well as balloon platforms and ground-based instruments. The mission also acquired correlative data needed to validate the Stratospheric Aerosol and Gas Experiment (SAGE) III satellite measurements that is used to quantitatively assess high-latitude ozone loss.

The SOLVE campaign sought to establish a comprehensive data set of UTLS trace gases
and meteorological data in the northern polar regions, including latitudinal gradients across the polar vortex. The campaign took place during the period October 1999 to March 2000. $CO_2$, $CH_4$ and $N_2O$ were measured by several instruments and used to calculate a composite AoA (Andrews et al., 2001b).

In the UTLS, there are strong horizontal and vertical gradients. These gradients may occur
on a small scale and show a high temporal and spatial variability. In this study, we were interested in airborne measurements to characterize the stratospheric variability and mixing process as well as to validate our model. The aircraft atmospheric measurements have a huge advantage for the spatial resolution and offer the possibility to capture with high precision small scale variations. Aircraft observations have a vertical resolution of a few meters and a horizontal reso-
lution of a few hundred meters resulting from the high sampling frequency of these instruments (0.5-2 Hz). Test and transit flights, made at mid-latitudes, were used to illustrate the comparison of the calculated $CO_2$.

*In situ* balloon-based $CO_2$ profile measurements are used as basis for comparisons with the reconstructed $CO_2$ from the Lagrangian transport model. The data sets are high-quality ob-
servations with sufficient altitude coverage and are from measurements of whole air samples collected cryogenically from balloons or from *in situ* measurements on-board a balloon gondola (Engel et al., 2009; Ray et al., 2014). Four balloon flights covering the time period from 2001 to 2005 and the latitudes range from 34 ° to 44 °N with a vertical resolution of a few me-





ters are used in this study. Only these four balloon flights contained the $CO_2$ profiles that were measured at Ft. Sumner, New Mexico, USA (34.5 °N) on 17 September 2004, Sanriku, Japan (39.33 °N) on 30 May 2001, Aire sur L'Adour, France (43.5 °N and 44 °N) on 24 September 2002 and on 9 October 2001, respectively. Note that most profile observations are from the May to October period, when stratospheric variability in the northern hemisphere is expected to be lower than during the winter period. These measurements of very long-lived trace gases, which increase with time in the troposphere and have neither atmospheric sinks nor sources of significance in the relevant parts of the atmosphere, have been used to derived the mean AoA (Andrews et al., 2001a; Engel et al., 2009) but here we focus on $CO_2$.

## 4   Initialisation of air particles

In order to calculate the global $CO_2$ distribution, air particles are initialized from the UT to the stratosphere their backward integrations. This initialisation is described below.

### 4.1   $CO_2$ global initialisation

The global initialisation aims to obtain a zonal mean distribution of $CO_2$ in order to investigate the mechanism of the BDC over the whole UT/S. To perform this study so, we assign these air particles globally integrated in the Sect. 2 with the reconstructed free tropospheric $CO_2$ field in Sect. 3.1 over the UT and the whole stratosphere.

In the model the assignation of the reconstructed free tropospheric $CO_2$ field (Sect. 3.1) is performed according to latitudinal and longitudinal position of the air particle for a given time. Each trajectory is integrated backward to the troposphere and at that point we assign the free tropospheric $CO_2$ to it at that time based on wherever the trajectory reaches the tropospheric boundary condition. It is assumed that the vertical mixing is fast in the troposphere, which is verified in the inner tropical region, where are the sources of the stratospheric air (Brewer, 1949). As the air particle moves upward across the boundary level, it retains its last value of $CO_2$ inherited from the boundary condition. The $CO_2$ value of the air particle is then advected





into the stratosphere by the different branches of stratospheric circulation.

The monthly mean $CO_2$ for a given box in latitude and altitude (typically $2°\times$ model level spacing) and for a given month is calculated as the average in longitude circle over all particles falling within this box (Fig. 1). This calculation uses the same approach as Sect. 2.3 in Diallo et al. (2012). Owing to the quasi-uniform spread of the discrete trajectories at the initialisation stage, the average is made over 180 particles at the equator and over 67 particles at $68°$ N or S. Latitudes closer to the pole are grouped into enlarged latitude bins ($69°$–$73°$, $73°$–$77°$, $77°$–$81°$, $81°$–$90°$) to avoid large fluctuations due to the reduced number of particles. Further averaging over time is performed to improve statistics and to reduce noise. These averaging procedures are a simple way to account for mixing in the stratosphere and gather within each box a distribution of particles with different histories.

As observed by Scheele et al. (2005), the number of backward trajectories launched at a given date and remaining within the stratosphere after some delay age (e.g. time residence), $\tau$, decreases exponentially with $\tau$. Figure 1 in Diallo et al. (2012) shows that this law is indeed very well satisfied for $\tau > 3\,\mathrm{yr}$ with an exponential decrement $b = 0.2038\,\mathrm{yr}^{-1}$ for the mean decay in ERA-Interim and that the standard deviation from the mean (when each month is considered separately) decays at the same rate. After 10 years of backward motion, $88\,\%$ of the particles launched within the stratosphere have met the tropopause. We follow Scheele et al. (2005) in using this property to correct the estimated $CO_2$ for the truncation of trajectory lengths at 10 years. If we define $G(\tau,t)$ as the cumulative probability density of the age $\tau$, the monthly mean $CO_2$ is

$$\overline{CO}_2(t) = \int_0^\infty CO_2(t-\tau) \cdot G(t,\tau) d\tau. \tag{1}$$

The truncated version of this integral, up to $t_f = 10\,\mathrm{yr}$, can be calculated explicitly from the backward trajectories. Assuming that $G(t,\tau) = G(t,t_f) \cdot \exp\left(-b(\tau - t_f)\right)$ for $t > t_f$ with $CO_2(\tau) = p_0 + p_1 \cdot \tau + b_0 \cdot \cos\left(2\pi(\tau - \varphi)\right)$ obtained by fitting the Mauna Loa $CO_2$ data, the monthly mean $CO_2$ can be estimated as





$$\overline{CO}_2(t) = \int_0^{t_f} CO_2(t-\tau) \cdot G(t,\tau) d\tau + \frac{G(t,t_f)}{b}\left(p_0 + p_1 \cdot \left(t - t_f - \frac{1}{b}\right)\right) + \text{Corr}(t), \quad (2)$$

with

$$\text{Corr}(t) = \frac{b_0 \cdot G(t,t_f)}{b^2 + 4\pi^2}\left[b \cdot \cos\left(2\pi(t - t_f - \varphi)\right) + 2\pi \cdot \sin\left(2\pi(t - t_f - \varphi)\right)\right]. \quad (3)$$

The obtained zonal mean distribution of $CO_2$ is further discussed below but currently we focus on the reconstructed $CO_2$ for the SOLVE campaign in order to test the ability of the Lagrangian model to reproduce the aircraft measurements.

## 4.2 Validations of the technique

The *in situ* measurement location initializations during SOLVE and balloon campaigns are used for validation of the global initialisation of air particles.

### 4.2.1 SOLVE flight track initialisation

The ER-2 flight initialisation is made by using the airborne measurements during the SOLVE campaign. The particles are initialized along the flight trajectory with a frequency identical to that of the tracer measurements (0.25 Hz). For this calculation, a set of 200 particles are initialised at the same position as those measured during the flight in order to take into account the diffusive processes e.g. each air parcel is actually a mixture of particles from various origins. These particles are released every 4 s and integrated backward over 6 months with a diffusivity parameter $\kappa$ equal to $0.1\,\mathrm{m^2 s^{-1}}$ that represents the turbulent diffusion $D$ (Eq: 6) due to small-scale motion missing in the ECMWF winds (Legras et al., 2005). This procedure is chosen because it is consistent with the underlying physics and also because the applied diffusion can be estimated from the comparison of small-scale observed tracer fluctuations and reconstructions (Legras et al., 2005; Pisso and Legras, 2008). After a few days, the notion of isolated air parcel in the atmosphere loses its consistency. This model is a discretised approximation of the Green function obtained by integrating the adjoint advection-diffusion equation backward in time.



The dispersion of particles is done by adding a random vertical noise into the calculation of each trajectory and at each time step representing the turbulent diffusion $\kappa$ due to small scale movements not taken into account in wind fields (Legras et al., 2003). The displacement $X$ of air particles over a small time interval $\delta t$, is described by

$$\delta \mathbf{X} = \upsilon(\mathbf{X},\mathbf{t}).\delta \mathbf{t} + \delta \eta(\mathbf{t}).\mathbf{k} \tag{4}$$

where the white noise process for the mixing is $\delta \eta(t) \equiv w \delta t$ and $\mathbf{k}$ is the unit vector. The process is without memory (i.e it is $\delta$-correlated), and with a zero mean. If the limit $\delta t \longrightarrow 0$ and after statistical average over a large number of particles, this is equivalent to adding a diffusion to transport such the concentration tracer, $C$, is no longer conserved. Then advection-diffusion equation is

$$\frac{\partial C}{\partial t} + \upsilon.\nabla C = \kappa \frac{\partial^2 C}{\partial \theta^2}, \tag{5}$$

with

$$\kappa = \frac{1}{2} <w^2> \delta t \tag{6}$$

$w$ is derived from a white noise normalized by the value of the diffusion coefficient $\kappa$ defined by the user and $\delta t$ is the time step. In order to ensure that vertical velocities are bounded, we use a white noise based on a random variable $r$ that is uniformly distributed with zero mean and unite variance. Applying 6, the random process is then $\delta \eta = r\sqrt{2\kappa\delta t}$, where $\delta \eta$ is the vertical displacement with a new drawing of $r$ at each time step and for each particles. A corresponding displacement in $\theta$ is applied for diabatic runs using the adiabatic conversion function

$$\frac{\delta \theta}{\delta \eta} = \left(\frac{p_o}{p}\right)^\kappa \frac{g}{C_p}. \tag{7}$$

After the backward integration of few months, then the air particles of the diffusive run are fixed by using the ages calculated from the global initialisation.



### 4.2.2 Balloon flight initialisation

The balloon flight initialisation is made by using the *in situ* measurements campaign described in Engel et al. (2009). The particles are initialized along the balloon flight trajectory with a frequency higher than the tracer measurements. A set of 5000 particles are initialised over 200 vertical levels ranging from $500\,\mathrm{hPa}$ to $1\,\mathrm{hPa}$ at the same latitude-longitude position as those measured by the balloon flights. These particles are released as the balloon moves upward and integrated backward over 6 months with a diffusivity parameter $\kappa$ equal to $0.1\ \mathrm{m^2.s^{-1}}$ (Legras et al., 2005). After a few days, because of the consistency lost due to the notion of non-isolated air parcel in the atmosphere, we apply a same technique for the dispersion of particles. The mean $CO_2$ is obtained by constraining these backward trajectories with those calculated by the global initialisation.

## 5 Reconstructed $CO_2$

In order to determine how tracer distribution is controlled by transport and small scale of mixing into the extra-tropical LMS, we investigate how Lagrangian diffusive reconstructions (Legras et al., 2005; James and Legras, 2009) are able to reproduce the observed $CO_2$ (Andrews et al., 2001b; Daube et al., 2002; Park et al., 2007a; Engel et al., 2009; Ray et al., 2014).

### 5.1 SOLVE campaign

Here we reconstruct the $CO_2$ measured from the ER-2 flights in the polar regions during the SOLVE campaign using backward trajectories. The reconstructed $CO_2$ is obtained by averaging the $CO_2$ values at the locations reached by the air particles integrated in Sect. 4.2.1 after the end of the backward integration. These reconstructions exhibit a good agreement with the observations (red) from Daube et al. (2002). In all cases, the measurements fall within the 95% confidence interval of the reconstruction and in many instances reproduce the small scale variations as well. The large peaks seen, e.g., on 20000202, 20000226, 20000131 are due to dives and the discontinuities on 20000203 and 20000114 are also due to fast altitude changes.



However, on 20000127 and 20000311 the sharp discontinuities of $CO_2$ are due to the repeated crossing of the vortex edge and peeled off filaments outside the edge (Konopka et al., 2003; Legras et al., 2005). In some cases, like 20000203, the confidence interval gets larger and the agreements is lower because the flight stayed within a region of high sensitivity to trajectory fluctuations (near an hyperbolic point, see Koh and Legras (2002)) due to a vortex splitting event. On 20000307, the flight occurred in the ageing vortex core. The too low prediction of TRACZILLA might be due to the large contribution of the corrective step (see section 4.1) in this instance.

The overall conclusion of this section is that reconstructed $CO_2$ by TRACZILLA, based on advection with resolved winds and a simple parameterisation of mixing by diffusion, explains very well the measurements made by the ER-2 aircraft during the Arctic SOLVE campaign.

## 5.2 Balloon vertical profiles

It is also interesting to perform a detailed comparison with balloon soundings which span a larger vertical range than the aircrafts. Such a comparison of the vertical profiles of the reconstructed mean $CO_2$ by TRACZILLA with those derived from observations of four middle stratospheric balloon flights (Engel et al., 2009; Ray et al., 2014) is shown in Fig. 3.

For three of the cases, most of the measurements fall within the 95% confidence interval of the reconstructed profiles and the local maxima at $23\,\mathrm{km}$ and $18\,\mathrm{km}$ on 20020924 and 20050901 are well reproduced. The reconstructed profile on 20011009 is significantly different from the observed profile. The balloon was launched from Aire sur l'Adour (France) while the area was crossed by a cold front associated with strong tracer gradient in the upper atmosphere (see the insert in Fig. 3). In such situations, the errors of the reanalysis which are not accounted in our statistical test might be large enough to explain the shift.

The reconstructed vertical profiles of the mean $CO_2$ is in very good agreement with all the *in situ* $CO_2$ measurements at all altitudes, even in the small scale variations occurring in the $15$–$25\,$ km bin except in the profile at $44\,^\circ\mathrm{N}$, which exhibits a spread behavior (Fig. 3). The mean *in situ* $CO_2$ from observations is much more spread in the high latitude profiles ($44\,^\circ\mathrm{N}$) above $15\,\mathrm{km}$. There is not a clear explanation about these observed fluctuations on the *in situ*



$CO_2$ profile.

Figure 3 also illustrate the increase of the stratospheric $CO_2$ abundance from 2001 (upper left) to 2005 (lower right). Again, the reconstructions show the ability of TRACZILLA to capture the features in the vertical propagation of the tracer variations even in the small scales.

## 5.3 Temporal series

To obtain additional accurate details about the upward propagation of the tropospheric $CO_2$ seasonal cycle into the LMS, we compare the temporal evolution of the monthly mean $CO_2$ from the global initialisation (Sect. 4.1) calculated by TRACZILLA with those derived from observations over several months. We choose the latitudes ranges 10 °S–20 °N and 50–60 °N to validate the tropospheric and stratospheric boundary conditions for $CO_2$ time series from TRACZILLA.

The top panel of Fig. 4 shows the comparison of the time series of modelled monthly mean $CO_2$ with those from CONTRAIL (Sawa et al., 2008, 2012) in the 10 °S–20 °N at 8–9 km altitude bins over the same time period August 2006 to January 2010. The comparisons are generally in good agreement with observations, which shows the ability of the model to capture the tropospheric $CO_2$ seasonal variation and the validation of tropospheric boundary condition.

The mid panel of Fig. 4 compares the modelled monthly mean $CO_2$ time series in a layer at 16–17 km and between 10 S and 20 N just below the tropical tropopause with the average of ground-based $CO_2$ data from Mauna Loa (19 °N) and American Samoa (14 °S) delayed by 15 days. We find, consistently with Boering et al. (1996) and Andrews et al. (1999, 2001a,b), that the amplitude of the two signals is the same but our delay is shorter than the two months found in previous studies. We recover a two-month delay at higher altitude in the layer 18–19km (not shown). The origin of this discrepancy is unclear but is perhaps due to the fact that previous studies merge measurements in the deep tropics and the subtropics.

The bottom panel of Fig. 4 shows the modelled monthly mean $CO_2$ in the latitude bin 50–60 °N at different altitudes ranging from 7–13 km over the same time period November 2005 to January 2010. These curves are compared with CONTRAIL measurements in the same latitude band (Sawa et al., 2008). The modelled and measured $CO_2$ differ generally by less than 1 ppmv





except a few isolated months like March 2006 and March 2009. Notice also the large deviation of the 12-13 km CONTRAIL measurement in April 2007. There is a shift of the order of 4–6 months of the mean $CO_2$ seasonal cycle above 11–12 km with respect from the tropospheric signal. This is due to the delay induced by the shallow branch of the BDC also found by Bönisch et al. (2009) and Sawa et al. (2008). The large discrepancies concentrated during the spring season suggest that large gradients of $CO_2$ span the measurement are in this period, as it will be confirmed shortly.

## 6   Global distribution of the zonal mean $CO_2$

The zonal mean distribution of $CO_2$ illustrates the main features of the BDC, such as mixing and transport variabilities through temporal and spatial evolution. Figure 5 shows the typical seasonal variation of the monthly mean $CO_2$ in latitude-altitude cross sections calculated by the Lagrangian transport model for 2010.

### 6.1   Upper-troposphere and Lowermost stratosphere

The zonal mean distribution of $CO_2$ in the free atmosphere, especially above 5 km, is driven by the large-scale transport processes. Fast quasi-isentropic mixing is combined with upwelling in the tropics and downwelling in the extra-tropical lowermost stratosphere. Figure 5 shows that, in the northern hemisphere, the tropospheric monthly mean $CO_2$ is dominated by a strong seasonal cycle in the UT reflecting the activity (breathing of photosynthesis and respiration). $CO_2$ concentration grow during fall and winter to reach a maximum in April-May followed by a rapid decay due to the spring biospheric bloom and reach a minimum in July-August. The cycle is much weaker in the Southern hemisphere and influenced by the transport from the northern hemisphere. The combined effect of fast isentropic mixing (Haynes and Shuckburgh, 2000a,b) and convection (Sawa et al., 2008) propagates the cycle towards the tropics creating both horizontal and vertical gradients (Nakamura et al., 1991; Bönisch et al., 2009; Sawa et al., 2012). It is visible in Fig. 5 that during the northern hemisphere winter the concentration tends





to follow the isentropes in the extra-tropics. The barrier effect of the subtropical jet (Miyazaki et al., 2009) generates a strong meridional gradient near 30 °N, which is maximum near 350 K. Once it has reached the tropic, $CO_2$ is then transported upward by tropical convection and propagated to the stratosphere through the BDC. Throughout the spring and summer, while the tropospheric $CO_2$ is pumped out by the biosphere, a layer of high $CO_2$ extends from the tropics to the northern mid-latitudes in the lower stratopshere due to the lower branch of the BDC (Bönisch et al., 2008). This transport is favored by the Asian monsoon anticyclone, which traps young air lofted from the surface and induces a flux to the extra-tropical stratosphere on its West side as it is eroded across the jet (Dethof et al., 2000; Bannister et al., 2004; Park et al., 2007b, 2008, 2009, 2010; Randel et al., 2010). Due to the time delay of this transport, the maximum concentration of $CO_2$ in the northern lower stratosphere lags by 4 to 6 months that at the surface and is reached basically when the surface concentration is at its minimum. The result is an inverted vertical profile, which is amximum in July and persists over the summer. A qualitative comparison between the reconstructed $CO_2$ in Fig. 5 and observation from Sawa et al. (2008) (see their figure 7) exhibits a good agreement in the cycle of the tropospheric and lower stratospheric $CO_2$ and in particular in the cycle of the inversion. There are, however, differences in the location and intensity of the meridional gradient, which might be due to the specific sampling of Sawa et al. (2008) that gives a strong weight to the most intense region of the Pacific jet stream.

## 6.2 Middle and upper Stratosphere

As the tropospheric seasonal cycle is transported into the middle and upper stratosphere through the *tropical pipe* and mid-latitude tropopause crossing, its amplitude decreases upward because of the combined effect of the upwelling branch of the BDC and mixing processes. The deep branch of the BDC is much slower than the shallow branch and old air with low $CO_2$ concentrates in the middle and upper stratosphere. The tropical pipe isolate young air with high $CO_2$, an effect, which is maximum during northern hemisphere winter in agreement with age of air calculations (Li et al., 2012; Diallo et al., 2012). The horizontal mixing homogeneizes $CO_2$ in the mid and high latitudes. Because of this prior mixing the winter containement within the po-

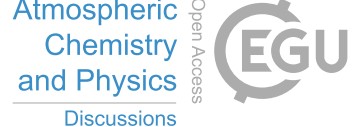

lar vortex does generates only a weak polar minimum (and no localized gradient as we average over latitude circle and not following the $CO_2$ contours).

## 6.3 Spring-Summer vertical profiles

In this section, monthly averaged $CO_2$ profiles are investigated to understand the changes in the
$CO_2$ vertical structure within the UT/S (Fig. 6).

On the left panel, the spring-summer reconstructed vertical profiles of the $CO_2$ compared with those from CONTRAIL aircraft measurements for the year 2007 at 50–60 °N show a good tropospheric agreement and an inversion of the $CO_2$ vertical gradient during August in the LMS. The monthly mean $CO_2$ vertical profiles, calculated by backward trajectories, exhibits a complex vertical structure with gradient layers interspelled with no gradient layers.

The annual structure of the profile is made apparent in the right panel where we show averaged monthly profiles over the period 2000-2010 after removal of the mean $CO_2$ trend at each level. Starting from January, the increase of $CO_2$ in the troposphere penetrates upward in the stratosphere over the limited vertical range of the Extra-tropical Transition Layer (Hegglin et al., 2010; Gettelman et al., 2011) (ExTL), that is over 2 to 3 km above the tropopause as visible on the March profile. Between March and May, another processus occurs, which inject young air rich in $CO_2$ above 13 km. This can only be due to a tropical intrusion favored by the weakening of the tropical barrier at the end of the winter. The profile suggests that the intrusion is deep from 13 to about 23 km, that the well-mixed layer between 13 and 16 km is inherited from the well-mixed tropical tropospheric profile at such altitudes and that the mixing layer between 16 and 23 km is also inherited from the tropical lower stratosphere vertical gradient. The mixing layer persists with the same slope during the whole summer and the bottom of the intrusion corresponds to the maximum of $CO_2$ when the inversion is at its maximum. During fall, when the subtropical barrier is re-established, the gradient weakens, the residual well-mixed layer disappears and the profile returns to the fairly uniform shape of January.



## 7    Conclusions

We have reconstructed a zonal mean distribution of $CO_2$ at global scale ranging from the upper-troposphere to stratosphere using Lagrangian trajectories guided by reanalysed winds and heating rates from the ERA-Interim. This study is based on 22 years of data and provides a monthly mean $CO_2$ over the last 11 years (2000–2010). We have also performed direct reconstruction of different *in situ* campaigns, such as SOLVE, stratospheric balloons campaigns and CONTRAIL, that has been compared with these observations to validate the reconstructed $CO_2$.

Our reconstructed $CO_2$ for the ER-2 aircraft flight tracks as well as for the vertical balloon profiles and time series using TRACZILLA show remarkable agreement even in the small scale variations with *in situ* observations from SOLVE, stratospheric balloons and CONTRAIL campaigns, respectively. This demonstrates that this Lagrangian model is able to reproduce most of the observed transport and mixing variability along the aircraft flight track, to capture the features in the vertical propagation of the tracer variations and to represent well the tropospheric and stratospheric boundary conditions. This reconstruction suggests that the distribution of long-lived tracers, such as $CO_2$, can be explained by the properties of transport as resolved by meteorological analysis and a simple representation of sub-grid scale effects as a diffusion.

The zonal mean distribution of $CO_2$ accurately reproduces the main feature of the stratospheric circulation variability through the upward and meridional propagation of the seasonal cycle extending from the upper-troposphere to the stratosphere.

In the northern hemisphere troposphere, the monthly mean $CO_2$ is dominated by biospheric activity and displays a strong seasonal cycle, which is vertically and horizontally propagated to the tropopause and above in the ExTL on the one hand and to the tropics on the other hand. In the regions of high horizontal mixed, as the mid-latitudes the $C0_2$ tends to uniformise over isentropic surfaces. Meridional gradients are localize over transport barrier such as the subtropical jet during winter.

Transport of $CO_2$ to the northern extra-tropical stratosphere above the ExTL is to a large extend due to the export of tropical air. The long circuit of $CO_2$ from the extra-tropics to the tropics in the tropopshere and then back to the extra-tropics in the stratosphere induces a time lag

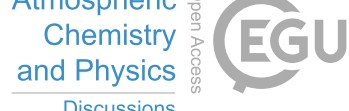



of 4-6 months such that the tropospheric and stratospheric varaibility are almost in opposition at mid-altitudes. The result is the production of an inverted $CO_2$ profile during summer.

In the deep stratosphere, we have found that as the tropospheric seasonal cycle is transported into the stratosphere through the *tropical pipe* and mid-latitude tropopause crossing, its amplitude is smoothed out upward the stratosphere because of the combined effect of the upwelling branch of the BDC and quasi-horizontal mixing. A more confined *tropical pipe* is found in the tropical band during the winter and spring than during summer and autumn.

Finally, the zonal mean distribution of $CO_2$ is a unique data set of a critical trace gas, that has a variety of uses. We have compiled this new relevant database of $CO_2$ global distribution from 2000–2010 for use to validate the representation of upper-tropospheric and stratospheric tracer distributions in CTMs, in CCMs and to test the representativeness of aircraft Measurements. The data set includes monthly-averaged two-dimensional fields gridded on the model's 77 latitudes and 36 vertical levels varying from $90\,^{\circ}$ S to $90\,^{\circ}$ N and from $5\,\mathrm{km}$ to $42\,\mathrm{km}$ respectively.

*Acknowledgements.* We particularly thank Y. Sawa, H. Matsueda and T. Machida for providing $CO_2$ measurements from CONTRAIL, the ECWMF for providing reanalysis data, WD-CGG for providing the ground station $CO_2$ data from 1989–1999, CarbonTracker team for providing the $CO_2$ data from 2000–2010. We thank Harald Bonisch for comments and helpful discussions. We acknowledge support from the EU $7^{th}$ framework Program under grant 603557 (StratoClim). M. Diallo was mainly supported by a postdoctoral grant from the ExCirEs Project (CGL2011-24826) funded by the Government of Spain. Further thanks to the CICLAD cluster at the Institut Pierre Simon Laplace in Paris on which most parts of this work had been carried out. The TRACZILLA model $CO_2$ data set may be requested from the corresponding author (mdiallo@lmd.ens.fr).

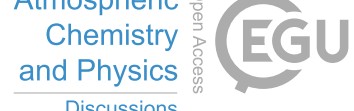



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



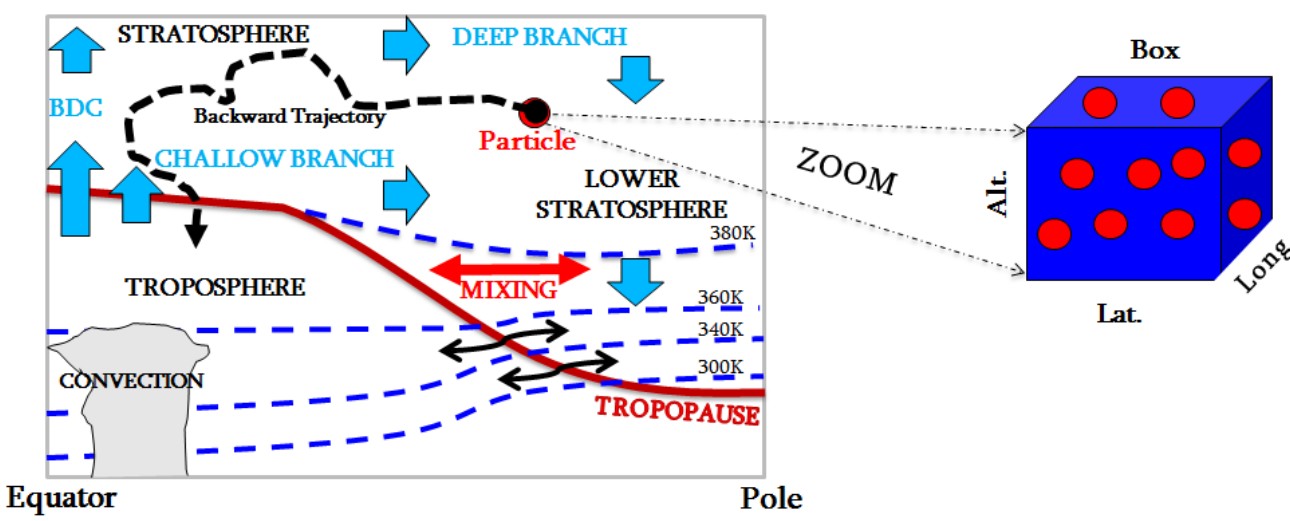

**Fig. 1**: A schematic representation of the initialization of air particles in the (latitude × longitude × altitude) grid box for the integration of backward trajectories.

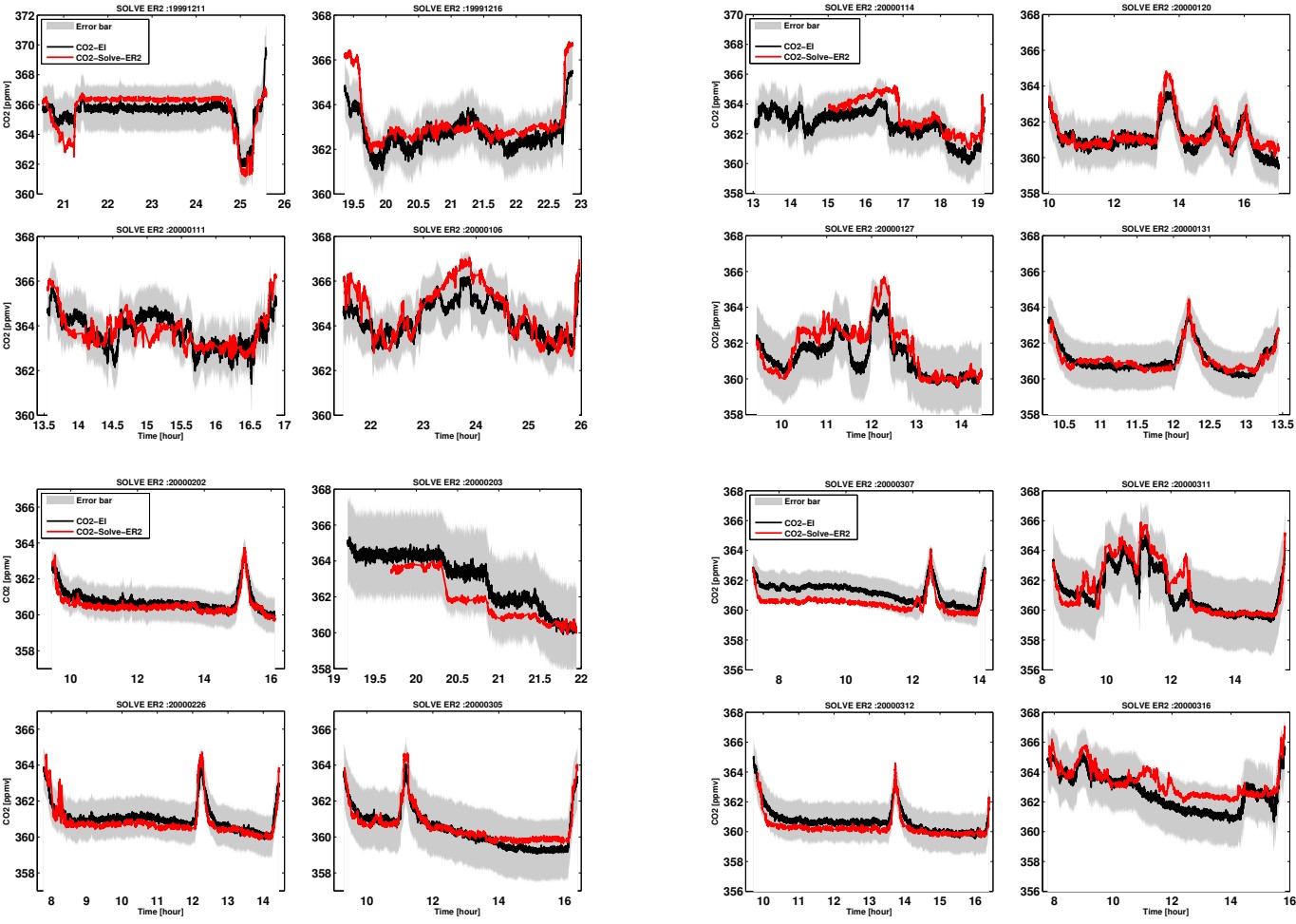

**Fig. 2**: Comparison between $CO_2$ calculated from backward trajectories and those from aircraft measurements during SOLVE campaign (Daube et al., 2002). (Black): reconstructed $CO_2$ alone the ER-2 flight track by the Lagrangian transport model. (Red): observed $CO_2$.





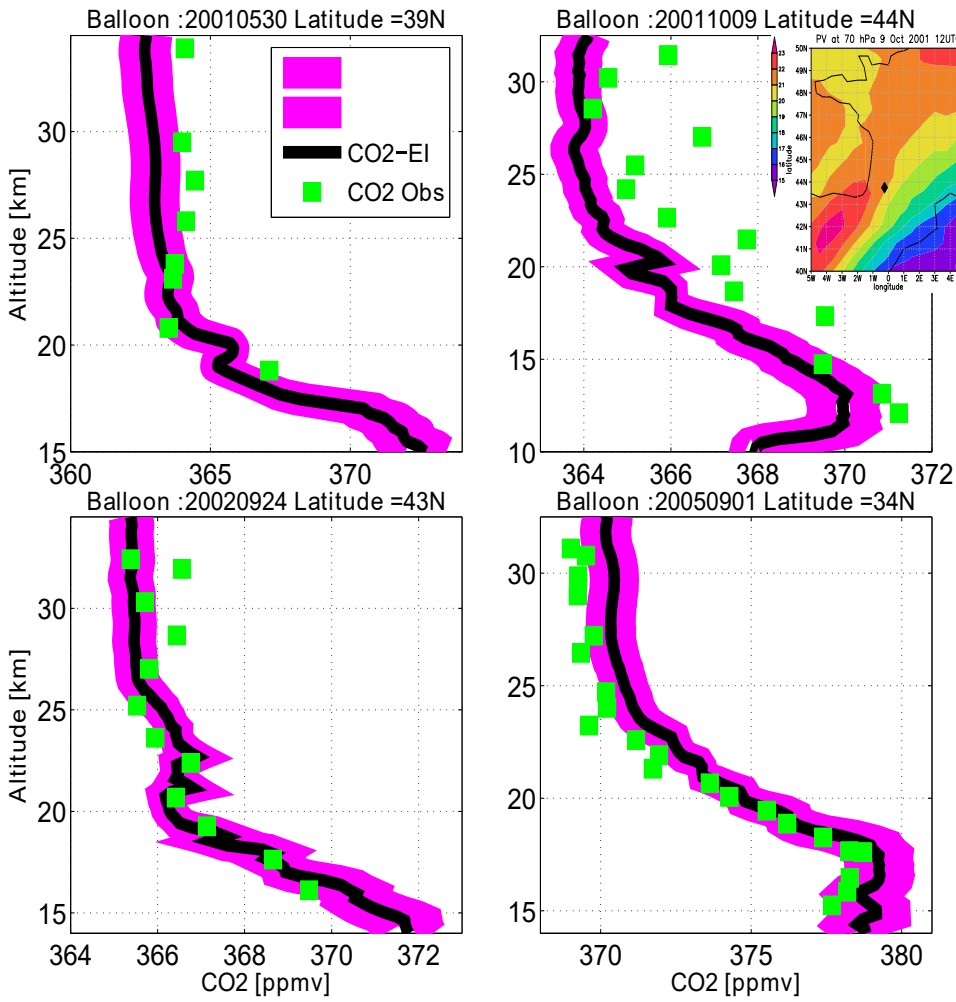

**Fig. 3**: Reconstructed vertical profiles of the mean $CO_2$ compared with those from each *in situ* stratospheric balloon observations of $CO_2$ (Engel et al., 2009). (Black curves): vertical profiles of mean diabatic $CO_2$. (Green square): *in situ* balloon measurements of $CO_2$. The measurements were taken from Sanriku, Japan (39.33 °N) on 30 May 2001, Aire sur l'Adour, France (43.75 °N) on 24 September 2002 and on 9 October 2001 and Ft. Sumner, New Mexico, USA (34.5 °N) on 17 September 2004, respectively. The insert on the upper right panel shows the potential vorticity (in PVU) on the 70 hPa surface for 9 October 2001 at 12UTC over France from ERA-Interim. The location of Aire sur l'Adour is indicated by a diamond.



**Fig. 4**: Evolution of the monthly mean $CO_2$ seasonal cycle from TRACZILLA calculations (line) compared with those derived from CONTRAIL measurements (circle). (Top): Tropospheric boundary condition for TRACZILLA $CO_2$ averaged from $10\,°S$–$20\,°N$ and compared with CONTRAIL measurements for the same time period and latitude bin. (Mid): Stratospheric boundary condition for TRACZILLA $CO_2$ averaged from $10\,°S$–$20\,°N$ and compared with the average of surface station data from Mauna Loa, Hawaii ($19\,°N$) and American Samoa ($14\,°S$) delayed by 15 days (Andrews et al., 2001a). (Bottom): Comparison of the $CO_2$ seasonal cycle from TRACZILLA with CONTRAIL near extra-tropical tropopause at 50–60 °N and at several heights from 7 to 15 km over the same time period from November 2005 to January 2010 (see Sawa et al. (2008) for CONTRAIL data).





**Fig. 5**: Global distribution of the seasonal cycle of the mean $CO_2$ (in ppmv) from the free troposphere to the stratosphere shown as latitude-altitude cross sections for January, March, May, July, September and November of the year 2010. The white lines indicate the isentropic surface levels for each given month.





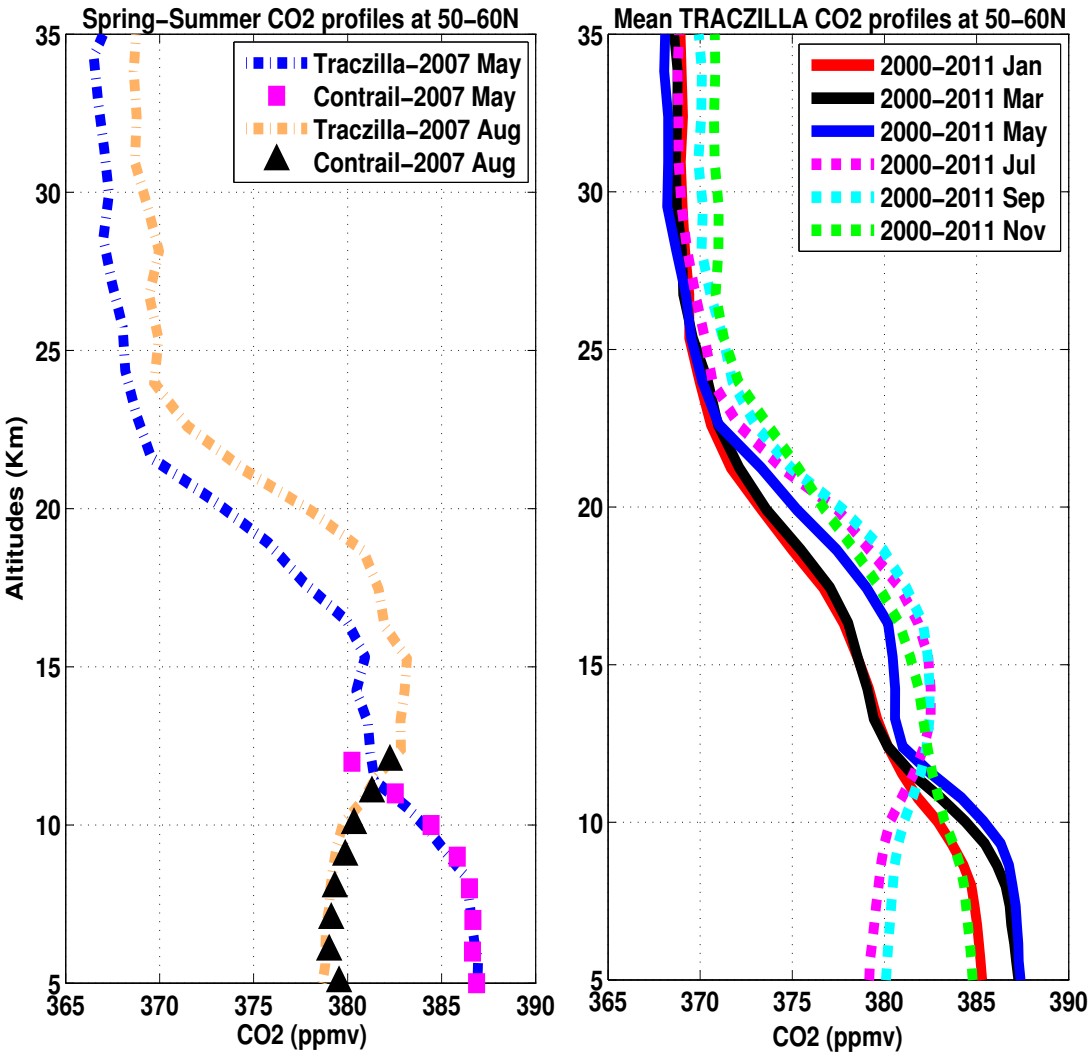

**Fig. 6**: Reconstructed vertical profiles of the mean $CO_2$ compared with those from CONTRAIL aircraft measurements for 2007 at 50-60 °N (Left). Averaged monthly profiles of the reconstructed $CO_2$ over the period 2000-2010 after removal of the mean $CO_2$ trend at each level and centered on 2007. Left: (Dotted-dashed lines): vertical profiles of $CO_2$ from TRACZILLA (blue: May, orange: Aug). (Symbols): *in situ* aircraft measurements from CONTRAIL campaign (magenta square: May, orange triangle: Aug). Right: monthly vertical profiles from TRACZILLA (January (red), March (black), May (blue), July (magenta), September (cyan) and November (green) ).