# Peer review of "Global distribution of CO2 in the Upper-Troposphere and Stratosphere"

_Atmospheric Chemistry and Physics, 2016_

## Referee Comment (RC1) · Anonymous Referee #3 · 21 Jun 2016

**General:**

The paper presents the global Lagrangian reconstruction of CO2 in the upper troposphere and stratosphere for the time period 2000-2010. This reconstruction is validated with in situ observations. It is astonishing that such a simple method can reconstruct CO2 with such a quality. Nevertheless there are still some differences and a possible origin of these differences is not well discussed (representation of convection, mixing, etc.)

The monthly zonal means are used for the description of the seasonality of transport both in the troposphere and in the stratosphere as well for the interpretation of the BD circulation. The results show a very impressive, detailed and quantitative picture how the seasonal cycle of CO2 ("breathing of the Earth") propagates upwards into the stratosphere. This important contribution is supported by well-performed figures. The

formation of the inverse vertical gradients of $CO_2$ over the course of the year is a very interesting feature which can be used to validated other transport models.

However, the quality of the presentation, especially of the text can (and has to) be improved. I think that the very experienced co-authors could help to do this job. The paper may be acceptable after a major revision improving this point

**Major points:**

1. My only major point is the quality of the text (outline, titles of the sections and subsections). I am not a native speaker but my feeling is that not only the structure of the paper but also the quality of many sentences can be significantly improved.

**Minor points:**

1. Abstract L 1
   please remove "relevant". You can only hope that this will be a relevant data set

2. Abstract L 8
   I would replace "guided" by "driven"

3. Abstract L 11
   ...with mid-latitude vertical profiles measured in situ from aircraft and balloons exhibit a remarkable agreement...(you should not give a complete description of the used data in the abstract)

4. Abstract L 17
   ...out of the tropics to the mid and high latitude stratosphere (but mainly into the northern lowermost stratosphere around 15 km - is it not something that follows from your Fig. 6)

5. Abstract L 21
   ...and is nearly constant above 35 km (is it what you would like to say ?)

6. P.4 L3
   These studies.... - please rewrite this sentence

7. P.4 L12
   ...to help to validate the stratospheric representation in global CTMs...

8. P.4 L14
   ...to the very localized in situ observations which have high spatial resolution, a large spatial...

9. P.4 L18
   Chadin et al showed.... (please rewrite this sentence)

10. P.5 L5
    ...are also weak... (what do you mean ?)

11. P.5 L21/22
    "model's lack of realistic stratospheric influence" - not clear, please explain

12. P.6 L10
    The small scale variability....and the scarcity of suitable observations ...(I would recommend to reformulate this sentence)

13. P.7 L14
    Trajectory starting....(you are using backward trajectories, so maybe you would like to write: "Trajectories reaching the boundary layer during the backward integration..."

14. Section: Data
    You should shortly describe here the aim of both upcoming subsections

15. Section 4 and 4.1
    I think, you should use a different title like "Reconstruction of CO2" and describe

it accordingly. "Initialization" is a very misleading term. So you use backward trajectories plus Carbon Tracker/WDCGG data in the boundary layer to reconstruct $CO_2$ everywhere in the UT/S region. Please reformulate the text between L10 and L20....

16. P.11 L14
Maybe you should change $b_0$ (which is too close to $b$) to something different.

17. P.12 eq (2) and (3)
Maybe you should avoid to introduce $Corr$, i.e. use only one formula in two lines

18. P.11 eg. (1)
You also did not clearly explain that you need your eq. (1)-(3) only for $CO_2$ reconstruction cases which are older than 10 years and you do not have any information from the backward trajectories. Maybe you should reformulate some sentences...

19. P.12 4.2.1
Once again: for me it not a "flight track initialization" but much more a "Lagrangian reconstruction of $CO_2$ along the flight track" - maybe you should reformulate it

20. P.12-13
You repeat here many arguments and formulations from Lugs Ed al., 2005. I do not think this is necessary. I strongly recommend to remove this part and cite the original papers. Instead of this, you should better explain your eq (7), i.e. how displacement in the geometric space is related to a displacement in $\theta-$space.

21. P.14, 4.2.2
Once again, please use the term "Reconstruction" in the title of this section and change accordingly the following text...

22. P.14 5.1
    Here, I miss any reference to Fig 2.

23. P.14 L 24
    Your abbreviations of the dates are not clear and maybe you should the notation like 26th February 2000, etc.

24. P.15 L 6
    aging vortex core

25. P.15 L 7
    "corrective step in this instance" - I do not understand what you mean

26. P.16 L 6
    "accurate" - please remove it

27. P.16 L 8
    ...from the global reconstruction calculated by...

28. In Fig 4a and 4b you denote the regions as tropospheric and stratospheric boundaries. However, in the model there is only one lower boundary prescribed by the Carbon Tracker values. You should exactly say what you mean with your boundaries. For me, 4a validate your model in the middle troposphere around 7-9 km and 4b in the region between 16-17 km. Please clarify this point.

29. P.17 L 3
    with respect to

30. P.17 L 4-7
    please give a more detailed explanation of the discrepancies. "as it will be confirmed shortly" could be replaced by "We discuss this point in the next section".

31. P.17 L 12
    ...derived from our Lagrangian reconstruction

32. P.17 L 19
    grows

33. P.17 L 21
    in the southern hemisphere

34. P.18 L 4
    propagates

35. P.18 L 5
    is removed from the atmosphere due to...

36. P.18 L 6
    into the lower stratosphere driven by the lower branch...

37. P.18 L 9
    west side (can you give more detail how the Asian monsoon anticyclone contributes to this transport)

38. P.18 L 13
    which is maximum...

39. P.19 L 1-2
    localized gradient, etc. - please give a more detailed explanation
* * *

---

## Referee Comment (RC2) · Anonymous Referee #1 · 21 Jun 2016

Review of acp 2016-0382

Global distribution of CO2 in the upper-troposphere and stratosphere

by Diallo et al.

This paper presents a new data base of monthly zonal mean distribution of CO2 in the UTS on global scale. The subject of the paper is scientifically very relevant and the scientific approach and applied methods are in principle acceptable.

Although the paper contains some interesting material which should be published, the manuscript itself has poor quality in many parts. The preparation of the manuscript was not done with the necessary care. Several paragraphs and sections need substantial revisions! The structure of the paper is fair but the grammatically preparation is superficial and the discussion with regard to the scientific content is inadequate. Particularly Chapters 5 and 6 are unclear and both do not contain a detailed discussion of the data product. In the following here are my major points and general concerns:

- In Section 3.1, I am missing a critical discussion of the boundary conditions for your exercise, for instance regarding the used data (i.e. the 1989-1999 time period from ground stations): what are the uncertainties of the calculated trajectories (e.g. with respect to the linear interpolation along latitudes (page 8, line 15-16))? Why are the $CO_2$ values from the CarbonTracker considered only at 5 km altitude? Are the results of TRACZILLA affected by the assumed initial conditions? How reliable are the TM5 results in comparison to other CTMs? At least it would be necessary citing relevant literature to motivate your approach – this section does not show any reference.

- Regarding Chapter 4, I am missing at least a final statement (paragraph) discussing the uncertainties with respect to the final data product. For example regarding the point mentioned on page 13, lines 18-19: "After the backward integration of few months, then the air particles of the diffusive run are fixed by using the ages calculated from the global initialization." What does this mean? How reliable are these assumptions? What is the range of uncertainties here? Again, no reference is cited in this connection. Similar on page 14, line 7 you stated: "... with a diffusivity parameter k equal to 0.1 ...". It would be helpful to provide more information and a critical discussion of the assumptions used here.

- Chapter 5 is really poor. The description and explanation of results in Section 5.1 is very vague and need a substantial revision. This kind of evaluation (i.e. a comparison of colored lines) presented here is not sufficient. For example (page 14, line 24): "The large peaks seen, e.g. ... are due to dives and the discontinuities on ... are also due to fast altitude changes." It is difficult to understand what you are meaning. In particular the mentioned flights are not fitting together, some of them are missed ... and by the way, Figure 2 is not mentioned at all in Section 5.1. Figure 2 is showing some discrepancies? What are the specific reasons? What is the grey shaded area

standing for in Fig. 2? What is needed here is a detailed description of the data and a critical discussion of the strength and weaknesses. Your final statement (page 15, line 11) ". . . explains very well the measurements . . ." is not sufficient.

I have the same concerns with respect to the other sections in Chapter 5. In the beginning of Section 5.2 a detailed comparison is announced, but it is definitely not given in the following. It is still a description of pictures. And it is a mess: Four balloon flights are presented in Fig. 3, three of them are "showing" a good agreement (but only two of them are mentioned in the text; page 15, line 19); one shows significant differences!? The explanation at the end (lines 22-23) is " . . . the errors of the reanalysis which are not accounted in our statistical test might be large enough to explain the shift." This is definitely not an adequate explanation!

In Section 5.3 you said that you choose the latitude ranges 10S-20N and 50-60N. What is the reason for chosen these bins? Why are the latitudinal bins different? A more de-tailed discussion is necessary. Again a statement like "The origin of this discrepancy is unclear, but is perhaps due to the fact that previous studies . . .". By the way: which studies (no references are given)? And at the end of this section the sentence (page 17, lines 5-7) "The large discrepancies . . ., as it will be confirmed shortly." is not com-prehensible to me.

- What I have said in general with regard to Chapter 5 is also valid for Chapter 6. A more detailed discussion of the results and their evaluation are required. Some particular points with respect to Chapter 6: You are discussing results of the year 2010 only in Sections 6.1 and 6.2. What are the other 9 years showing (regarding annual cycle, variability, fluctuations, etc.)? Are they "identical" (similar) or different? Is 2010 exemplarily for the other years? Beyond that, no results are showed in Section 6.2 (not either in other sections) regarding the upper stratosphere (all figures show results at altitudes below 35 km). With regard to the Conclusions you stated (page 21, line 3): "In the deep stratosphere, we have found . . . ". This remark is not supported by investigations presented before.

- A separate discussion Chapter at the end is necessary, especially an assessment and classification of the data set regarding the current knowledge. In particular to point out the "new" findings and a rating about the quality of the provided data set would be helpful. As announced in the Introduction (page 6, line 21) "The global (!) distributions of $CO_2$ and its transport in the UTS (!) are investigated . . .". Therefore, it would be nice to see some "global" charts, covering the entire stratosphere in this paper: The final sentence of your manuscript (page 21, lines 12-13) makes me curious, but no respective illustrations are shown in this paper. Could you please show (and discuss!) some examples, not only to motivate using this comprehensive data set?

- Section 7 (Conclusions) provides mostly a brief summary of things which have been stated before.

Minor points:

- Many acronyms are not explains, in particular in the Introduction.

- Page 3, line 19: . . . increase global change . . .

- Page 4, line 14: . . .observations that have high resolution and are very localized, . . .

- Page 5, line 29 and in particular on Page 7, line 3: TRACZILLA is not explained (at least references are required).

- Page 10, lines 14-17: need a Revision

- Page 10, line 22: "It is assumed that the vertical mixing is fast . . ." Why? Reference is needed? How fast? Please explain.

- Page 14, line 7: Units should be corrected.

- Page 15, lines 15-16: ". . . of four middle latitude stratospheric balloon flights . . .".

- Page 20, lines 4-5: ". . . based on 22 years of data . . . over the last 11 years . . .". Sentence should be revised.

- Page 21, line 3: "In the deep stratosphere we have found . . . " no results are shown or discussed in detail.

- Figure 4: the chosen latitudinal bins do not exactly correspond to the coordinates of the stations providing the measurements.

- Figure 6: the box in the right figure (top right) says "2000-2011"; should be 2000-2010.

From my point of view this paper needs a substantial revision. I believe that many things have to be done before the paper can be published in ACP. The manuscript is far away from been ready. It definitely cannot be published as is. Of course, the results should be published, but the scientific content as presented, is too vague. The paper definitely needs especially a more critical explanation and discussion of the results; it is necessary to point out the strength and weaknesses (data quality) of the data set in more detail and to discuss the uncertainties. And finally, the quality of the presentation is in parts really poor. Only if the complete manuscript will be significantly revised, it might be considered in ACP; the revised version should be again get another round of reviews (2 reviews).

---

## Referee Comment (RC3) · Anonymous Referee #2 · 21 Jun 2016

The manuscript describes a method and a reconstruction of global CO2 mole fractions through the Upper troposphere and stratosphere. The CO2 reconstruction is based on a back-trajectory model that samples tropospheric mole fractions when the trajectories end at a boundary in the troposphere. The results compare very well with observations. The manuscript requires improvement: the methods are not clear enough (details are in the specific comments below), and, most importantly, there is no uncertainty esti- mate at all. This is vital if the product is going to be useful to other scientists. The authors should also indicate how this product will be available for public research use. In general, the methods seem sound, and the work should be published after major revisions.

General notes:

[Figure]

1) The global product requires uncertainty estimates/bounds and a description of that derivation. This is essential.

2) The manuscript needs to clarify the product for global CO2, that it is 2-dimensional (varies with latitude and altitude but not longitude). This was unclear earlier in the text, and only clarified in the final conclusion sentence. How this was generated in terms of where the receptors for the trajectories were distributed in the stratosphere is unclear. How are the initial air parcels distributed in time and space, and are they run from different longitudes and the results averaged in zonal means?

3) Grammar and spelling should be checked throughout the manuscript. I noted many of the errors below, but it should be thoroughly proofread. Many sentences are awkward. The quality of the writing seems to deteriorate even more in the final few pages, with basic typographical errors and usage of words that are not words.

Specific comments:

Overall, the journal editor can comment on whether italics are or are not appropriate where used in this manuscript, generally when defining a term or acronym (for example "tropical pipe" is italicized throughout the paper).

Abstract:

L13: the potential of [a, or the] Lagrangian model to reconstruct...

L18-19 enter should be enters

L21: decreases with altitude... is nearly constant with altitude - contradictory. Perhaps the authors intend that CO2 decreases with altitude from the UT to the S but is nearly constant with altitude above 35 km?

Page3: L3: should be "The increase of greenhouse gases, "

L14: seasonal cycle

[Figure]

L19-20 awk sentence (what is "its potential" referring to? - should be "their", in ref. to CO2 abundance and variability?).

L20 should be "abundance".

Page 4: L 6: where recently assessed from balloon-based

L 28, qualitative? Perhaps better worded would be "shown qualitatively good agreement with in situ observations..." Last line and L1 of Page 5, awkward sentence

Page 5:

L25 grammar

L18 this diffusivity effect

L29 models

I don't completely understand why Lagrangian models would not be subject to these problems if the underlying transport model is flawed.

Page 6: L10: by scarcity are the authors referring to the scarcity of CO2 observations? confusing sentence, possibly because of wording /grammar errors.

L15: is the ERA-interim analysis used in any of the previously referred-to flawed CTM models that were unable to correctly capture transport?

L15 ERA-Interim definition should be moved up here from Line 20-21.

Page 7: L2: the Lagrangian transport model TRACZILLA (Legras ..), a modified version....

L5: I see now that the Lagrangian model is calculating its own vertical motions, unlike perhaps the CTMs mentioned above, and that is why it performs better?

Line 15: Please clarify the assumption being made here: this corresponds to the assumption that the CO2 in the troposphere (i.e. below this boundary condition) is con-

[Figure]

stant?

Section 3.1, page 8. The use of clean-air data at the ground, no matter how clean the site, is a bit worrisome. Since Carbon tracker has also been used for the later years, how would it impact your results if you used a lower or higher CarbonTracker level? The 5km level is not only above the PBL, it is significatly higher than the other stations that you use in 1989-1999. It would be worthwhile to investigate the vertical gradient in CarbonTracker to see what kind of error you have just by choosing one level and assuming it is constant. Also it would be useful to look at CarbonTracker residuals against the NOAA North American aircraft network data at altitudes above the PBL, to see if it is a realistic depiction of CO2 mole fractions in the free troposphere and upper troposhere (those profiles go to 8 km).

http://www.esrl.noaa.gov/gmd/ccgg/carbontracker/profiles.php

If residuals are small, then using the CT gradient might be a good way to determine the uncertainty on the assumption and the choice of only one level. CT can also be used to investigate any longitudinal errors/differences.

It is not clear if the CarbonTracker mole fractions are considered as an average over all longitudes, or for the specific grid cell where the back-trajectory initiates.

L22-23: This is not a great description. CarbonTracker assimilates CO2 observations from atmospheric stations and optimizes underlying fluxes from the listed sources (ocean fluxes, biosphere fluxes, fire and fossil fuel).

Page 10: L19: I repeat my question from before, which could be addressed here - it is assigned based on the lat/lon of the boundary crossing after 2000 but prior to 2000 it is only the latitude that matters?

page 10-11: Are the particles in each bin spread evenly throughout? I.e. is their initial location in the center of the bin or evenly distributed in time and space? this may have been mentioned earlier but should be clarified here. (Page 11, line 6 discusses how

many particles there are per bin, so this would be a good place to discuss how they are distrubited in time over the month and in space).

Page 12, L10-15, this discussion of diffusion answers my previous question about the difference in the models - it could be briefly mentioned earlier in the paper to clarify this.

Page 13-14. Perhaps you could clarify why this 6-month dispersion of particles is required before the particles are tagged with the value of the global reconstruction. I would think you could just sample your global distribution at the location of the observations?

Page 14, Line 21, reference to figure needed in text here.

Page 14, L23: This is the first reference to a CI for the reconstruction. How is it calculated/obtained? The CI or uncertainty is very important to anyone who would be using this product, and the methods for its calculation should be well documented in this paper. Presumably some of this comes from the particle/diffusion release of the 6-month trajectories from the flight, but there should be an uncertainty associated with the global CO2 product as well.

Fig. 2: I would like to see differences in addition to the time series of mole fractions, or correlation plots with $R^2$, statistics related to bias and/or RMSE; i.e. a more quantitative comparison. A plot or a table of statistics, or perhaps a single plot with all the flights colored differently, would not add too much length or bulk to the paper.

Fig. 3 shows the differences well enough that it seems fine, and statistics would be difficult to calculate for such a small sample.

Page 15, L23: We have no information on the error calculation so this sentence is confusing.

Page 16, L 13: CONTRAIL should be described somewhere (in the section where SOLVE and the balloon flights are discussed in Section 3.2).

Page 16, section 5.3, why is CONTRAIL treated differently from Solve and the balloon observations, and no 6- month diffusion of particles is conducted? (this is the same as my earlier question about why that was done for those observations, there should be some explanation for this).

Fig 4, some uncertainty bounds should be calculated and shown for the global initialisation using TRACZILLA. (could be based on differences from Contrail, or based on the spread of values of the trajectories that constructed each bin). Some error is likely also introduced by the choice of the level in CarbonTracker (looking at any vertical gradients in CT product could help quantify this, perhaps it is very small).

Page 16: Why was 15 days chosen, was it because that delay fit the data best? This is significantly shorter than 2 months - perhaps this could be made more clear, that these are the two numbers being compared in the discussion in this paragraph (15 vs 60 days).

Page 16, last line: give mean, standard deviation of differences?

Page 17, Line 6-7: in this period, discussed further in section xX.

Page 17, Line 18: awkard description of the biospheric CO2 seasonal cycle

Page 17, L19: CO2 concentration in the UT increases (subject /verb agreement in this sentence).

Fig 5, caption should indicate that the source of this data is the CO2 global reconstruction

Page 18:

L3: tropics

L13 maximum

L14 observations, also refer to CONTRAIL here to clarify this is the same data set used

in the earlier comparisons

L25 isolates

L26 no comma after effect, also this is still awkward phrasing

L27 homogenizes

L28 containment

Page 19:

L9 profiles.... exhibit

L10 I don't think interspelled is a word, please rephrase (perhaps interspersed, or alternating?).

L16, processus (process)

L16, injects

L7 that should be which

L11 "This good agreement demonstrates that the Lagrangian model (TRACZILLA) ..."

L23-25: awk sentences (high horizontal mixed and uniformise)

L17 extent

L29 troposphere

L1 variability

L1 subject/verb (variability are)

L11 measurements should not be capitalized

---

## Referee Comment (RC4) · Anonymous Referee #4 · 28 Jun 2016

The authors want to provide a new data set of CO2 for model evaluation and the capabilities of models to simulate transport in the stratosphere. For this purpose they construct a CO2 data set on the basis of a Lagrangian approach (TRACZILLA) by developing a ten year zonal mean climatology of CO2 on the basis of ERA Interim data. They use Carbon Tracker CO2 fields in the free troposphere to build this climatology for the stratosphere. Comparisons to CO2 from the SOLVE campaign in winter 1999/2000 over the northern Arctic are presented. The model data are further compared to measurements of CO2 from the SOLVE and CONTRAIL project covering several years of CO2 measurements from commercial air liners mostly between Europe and Japan over mid-latitude and northern Asia. Four mid-latitude balloon-borne profiles allow for comparison of CO2 deeper in the stratosphere of which one shows considerable deviations to the model data, whereas the other show a remarkable agreement between model

and observations. The authors show that TRACZILLA is capable of partly reproducing the $CO_2$ variability also on small scales as provided by these data sets. Based on the few comparisons they claim to provide a new data set for model evaluation from the upper troposphere throughout the stratosphere for the years 2000 to 2010.

In general such a data set is highly desirable. However, in the presented form and with the methods used here, the validity of the $CO_2$ data set for such a task is not provided. The construction method seems to be valid, but the evaluation and interpretation of the results is confusing and based on limited data. Further, no statistical measures or quantitative metrics are used, which would really allow to draw conclusions for the general applicability of the data set as stated in the conclusions. The authors only provide model to observation comparisons for a few limited time series and profiles. Statistical robust evaluation methods are not applied at any point. Also the implications of inconsistencies to previous studies (e.g. the $CO_2$ tropopause condition) are not discussed, nor the reason for inconsistencies. If the data set presented is out of phase at the tropopause by 6 weeks, how is it possible to get the stratospheric $CO_2$-distribution correct?

The more important is a careful analysis of the stratospheric $CO_2$, where observations are sparse. Surprisingly they further don't calculate age of air, which should be easily possible with $CO_2$ at least at altitudes, where the seasonal cycle of $CO_2$ disappeared. I judge this as essential before claiming a reference data set for model evaluation.

I therefore think the paper should be resubmitted to ACPD, after 1) more quantitative statistical evaluation is provided 2) an age of air discussion and comparison (e.g. to MIPAS or balloon-borne data) is included to evaluate the stratospheric data 3) a thorough discussion and definition of processes and regions is given and used (e.g. the term 'tropical pipe')

A data set as intended by the authors would be of high interest for the community, but in the presented form it is just a $CO_2$ data set, which fits some selected and very limited

observations.

Major point: The evaluation of the data set is very specific and done on the basis of very limited and selected case studies: SOLVE only covers the high Arctic in winter 1999/2000. Similarly the four profiles are arbitrary snap shots, showing disagreement in one case, which is not even tried to be explained sufficiently (p.15).

Further the discussion of the data and differences to literature is very superficial: The time delay between the cycles is mentioned but not appropriately discussed, since Boering et al., 1996 use $N_2O$ = 310 ppbv as reference values to determine $CO_2$ at the tropopause, whereas in the manuscript a fixed altitude level is used, which is at or even below the tropical tropopause. This is not considered in the manuscript. Instead the authors conclude "...We recover a two-month delay at higher altitude in the layer 18–19km (not shown). The origin of this discrepancy is unclear but is perhaps due to the fact that previous studies merge measurements in the deep tropics and the subtropics." - This is not satisfying for a quantitative reference data set over 10 years.

Given the statements in the abstract (l.3-6: "...This product can be used for model and satellite validation in the UT/S, as a prior for inversion modelling and mainly to analyse a plausible feature of the stratospheric-tropospheric exchange as well as the stratospheric circulation and its variability..."), a quantitative assessment and evaluation also of the stratospheric data is required, e.g. by using age of air diagnostics. This would allow for comparison with other data sources or diagnostics (e.g Eyring et al., 2006; Haenel et al., 2015). For this the authors could also include SF6 to their analysis, since the authors conclude that their results hold for any long-lived species (p.20, l.14). Even if SF6 cannot be directly included, the age of air information can be inferred from the data. This would provide a quantitative comparison to evaluate the results on the basis of $CO_2$ and the consistency of the results within the model. It would further help to evaluate their stratospheric data using satellite observations of e.g. MIPAS SF6 in regions where the in-situ data are sparse or absent. Even without SF6 the calculation of age of air allows for comparison with other data sets.

Specific points: Abstract: Last sentence: Please clearify the sentence and specify: there's a contradicition: decrease or constant? Decrease of CO2 to 35 km or constant, constant with altitude above?

Introduction: Do you need the first sentence?

p.3, l.2-4: The increase of green house gases does not increase tropical upwelling mass flux, it is the effect on atmospheric temperature structure and wave propagation.

p.3, l.6: stratosphere instead of atmosphere? CO2 is destryed in the upper atmosphere.

p.4, l.7: Here you need to mention the Engel et al., (2009) study - not on the previous page (l.25), since it is not beased on airborne measurements.

p.7, l.2: You forgot 'TRACZILLA'

p.8, l.20-22 (and l.10 ff): What does this mean: Similarly to 1989-1999.... only at 5 km ? Please specify the altitude criterion of the selected stations for 1989-1999. How many stations contribute? It would be good to show a 3D distribution of the boundary condition: e.g. a zonal mean plot with latitude as y axis, time as x-axis and CO2 as iso surface above to see the gloabl distribution and allow for comparison with e.g. the NOAA CCGG data.

p.10, l.11: Wrong sentence? Something is missing...?

p.11, l.3: Is a zonal mean calculated? What are typical numbers of parcels per box?

p.12, l.4: Chapter title: The term 'validation' is used, but one can't validate the results, since you can have agreement for the wrong reasons. Therefore I suggest the term 'evaluation'.

p.12, l.15 ff.: The exponential factor b clearly depends on the driving data set. Does the exponential factor b further depend on the choice of the trajectory model and needs in principle to be determined for each individual trajectory model?

p.13, eqn.4/5: Please clarify the notation of vectors, scalar products and sclara quantities. Why is the 't' in bold font?

p.13, l.11: ".. kappa defined by the user..." Is k (kappa) chosen to have the same value in the whole atmosphere? Please add a word, which values have been selected or how a user has to define Kappa.

p.15, l.19-23: Please explain, how a cold front (which is a tropospheric feature) can affect the CO2 a 25 km altitude. The inset in Fig.3. is too small and does not contain a legend. It is further unclear, why the PV gradient should be associated with a CO2 gradient. This paragraph sounds very weird or almost wrong.

p.15, l.27-29: "... The mean in situ CO2 from observations is much more spread in the high latitude profiles (44 ◦ N) above 15 km. There is not a clear explanation about these observed fluctuations on the in situ CO2-profile.": What do the authors want to say with such a statement? What does this mean for the comparison? What does it mean for a data set, which is intended to serve as a reference for model evaluation from 2000-2010, if one out of four stratospheric profiles does not fit the observations?

p.19, l.10: What are gradient layers?

p.19, l.20 and l.23: What is meant with "the subtropical barrier" in this paper? Do you mean the subtropical jet at the tropopause, which exhibits a seasonality with weaker PV gradients and high permeability in summer? The subtropical barrier normally denotes the boundary of the (leaky) tropical pipe in the overworld (e.g. Palazzi et al., 2009), which does not show the same variability as the STJ and has a different generating mechanism.

Fig.2: Please include potential temperature along the flighttrack as additional information. Otherwise the information on the plots is without any relevance for a scientific interpretation and an estimate of the quality of the model capabilities. Does the gray area refer to the variability of the data in a bin? If not, how is the error calculated?

[Figure]

Fig.5: The continous color bar is not consistent with the figures, which have discrete colours. Please provide a discrete color bar legend.

References:

Boering, K. A., S. C. Wofsy, B. C. Daube, H. R. Schneider, M. Loewenstein, J. R. Podolske and T. J. Conway;Stratospheric Mean Ages and Transport Rates from Observations of Carbon Dioxide and Nitrous Oxide, Science 274 (5291), 1340-1343, doi: 10.1126/science.274.5291.1340, 1996.

Eyring, V., et al. (2006), Assessment of temperature, trace species, and ozone in chemistry-climate model simulations of the recent past, J. Geophys. Res., 111, D22308, doi:10.1029/2006JD007327, 2006.

Haenel, F. J., Stiller, G. P., von Clarmann, T., Funke, B., Eckert, E., Glatthor, N., Grabowski, U., Kellmann, S., Kiefer, M., Linden, A., and Reddmann, T.: Reassessment of MIPAS age of air trends and variability, Atmos. Chem. Phys., 15, 13161-13176, doi:10.5194/acp-15-13161-2015, 2015.

Strahan, S. E., Schoeberl, M. R., and Steenrod, S. D.: The impact of tropical recirculation on polar composition, Atmos. Chem. Phys., 9, 2471-2480, doi:10.5194/acp-9-2471-2009, 2009.

---

## Author Comment (AC1) · 29 Nov 2016

[acpd,hvmath]copernicus$_{discussionscolor}$

m.diallo@fz.juelich.de Answer to referee #1 Diallo, Legras, Ray, Engel and Anël

11

[Figure]
We thank referee #1 for his comments and suggestions. Comments by the referee are highlighted and followed by our answers.

Although the paper contains some interesting material which should be published, the manuscript itself has poor quality in many parts. The preparation of the manuscript was not done with the necessary care. Several paragraphs and sections need substantial revisions! The structure of the paper is fair but the grammatically preparation is superficial and the discussion with regard to the scientific content is inadequate. Particularly Chapters 5 and 6 are unclear and both do not contain a detailed discussion of the data product. In the following here are my major points and general concerns:

Major points:

1. ***In Section 3.1, I am missing a critical discussion of the boundary conditions for your exercise, for instance regarding the used data (i.e. the 1989-1999 time period from ground stations): what are the uncertainties of the calculated trajectories (e.g. with respect to the linear interpolation along latitudes (page 8, line 15-16))? Why are the $CO_2$ values from the Carbon-Tracker considered only at 5 km altitude? Are the results of TRACZILLA affected by the assumed initial conditions? How reliable are the TM5 results in comparison to other CTMs? At least it would be necessary citing relevant literature to motivate your approach – this section does not show any reference.***

   It's not straight forward to estimate of the initial condition in the 1989-1999 as there's no reference global distribution of the $CO_2$ concentration. The linear interpolation is meant to avoid artificial discontinuities in the assignment. The basic uncertainties in the trajectories arise from errors in the advection fields that can be sensitive in some limited circumstances, as illustrated in the comparison to balloons and aircraft measurements. It is proven (see Legras et al., 2005 and Diallo et al., 2011) that the reconstructions are weakly sensitive to the duration of the integration and to the small-scale patterns of the data used in the initialization. Only large-scale bias would be seen in the reconstructed $CO_2$. The CarbonTracker data are used as a boundary conditions at 5km as it is a convenient way to initialize the trajectories as they cross this surface which lays entirely in the free troposphere. The representation of vertical transport by diabatic heating and the mere usage of potential temperature as a vertical coordinate is highly unpractical closer to the surface. This is better handled by a tropospheric transport model like TM5, now properly referenced, but the details of which are highly irrelevant for our purpose. CarbonTracker data is mainly used as a low pass filter which damps the daily and local fluctuations of $CO_2$ at the surface and carries the

low frequency and large scale ground signal to the 500 hPa surface in a matter of days. A precision, we don't use TM5 data but the data from the assimilation system CarbonTracker which is coupled with TM5 for the transport.

2. ***Regarding Chapter 4, I am missing at least a final statement (paragraph) discussing the uncertainties with respect to the final data product. For example regarding the point mentioned on page 13, lines 18-19: "After the backward integration of few months, then the air particles of the diffusive run are fixed by using the ages calculated from the global initialization." What does this mean? How reliable are these assumptions? What is the range of uncertainties here? Again, no reference is cited in this connection. Similar on page 14, line 7 you stated: ". . . with a diffusivity parameter k equal to 0.1 ...". It would be helpful to provide more information and a critical discussion of the assumptions used here.***

This paragraph that contained an error has been rewritten. The method used here is based on previous works (Legras et al., 2005; Pisso and Legras, 2008) based on long-lived tracers which provide constrains on the diffusivity. We refer the reviewer to these previous works which contains a thorough discussion of the sensitivity of reconstructions to the diffusivity.

3. ***Chapter 5 is really poor. The description and explanation of results in Section 5.1 is very vague and need a substantial revision. This kind of evaluation (i.e. a comparison of colored lines) presented here is not sufficient. For example (page 14, line 24): "The large peaks seen, e.g. . . . are due to dives and the discontinuities on . . . are also due to fast altitude changes." It is difficult to understand what you are meaning. In particular the mentioned flights are not fitting together, some of them are missed . . . and by the way, Figure 2 is not mentioned at all in Section 5.1. Figure 2 is showing some discrepancies? What are the specific reasons? What is the grey shaded area standing for in Fig. 2? What is needed here is a detailed description***

*of the data and a critical discussion of the strength and weaknesses. Your final statement (page 15, line 11) "... explains very well the measurements ..." is not sufficient.*

This section has been heavily rewritten

4. *I have the same concerns with respect to the other sections in Chapter 5. In the beginning of Section 5.2 a detailed comparison is announced, but it is definitely not given in the following. It is still a description of pictures. And it is a mess: Four balloon flights are presented in Fig. 3, three of them are "showing" a good agreement (but only two of them are mentioned in the text; page 15, line 19); one shows significant differences!? The explanation at the end (lines 22-23) is " ... the errors of the reanalysis which are not accounted in our statistical test might be large enough to explain the shift." This is definitely not an adequate explanation!*

We have improved the discussion. The third case is associated with a meteorological front, a situation which is well known to amplify the advection errors as previously shown by, e.g., Pisso and Legras (2008).

5. *In Section 5.3 you said that you choose the latitude ranges 10S-20N and 50-60N. What is the reason for chosen these bins? Why are the latitudinal bins different? A more detailed discussion is necessary. Again a statement like "The origin of this discrepancy is unclear, but is perhaps due to the fact that previous studies . . .". By the way: which studies (no references are given)? And at the end of this section the sentence (page 17, lines 5-7) "The large discrepancies ..., as it will be confirmed shortly." is not comprehensible to me.*

The reason of the choice of the two latitude bands is to select two regions respectively representative of the tropics and of the extra-tropics. Then the altitude ranges are dictated by the location of the tropopause in these two regions, respectively at 10-11 km and at 17-18 km. We have improved the discussion and added references.

6. ***What I have said in general with regard to Chapter 5 is also valid for Chapter 6. A more detailed discussion of the results and their evaluation are required. Some particular points with respect to Chapter 6: You are discussing results of the year 2010 only in Sections 6.1 and 6.2. What are the other 9 years showing (regarding annual cycle, variability, fluctuations, etc.)? Are they "identical" (similar) or different? Is 2010 exemplary for the other years? Beyond that, no results are showed in Section 6.2 (not either in other sections) regarding the upper stratosphere (all figures show results at altitudes below 35 km). With regard to the Conclusions you stated (page 21, line 3): "In the deep stratosphere, we have found ... ". This remark is not supported by investigations presented before.***

The results of the year 2010 is considered here as exemplary for the other years. The annual cycle of the zonal mean $CO_2$ for the other 9 years look similar but differ to a growth rate that changes year to year. Due to this changing growth rate, the variability of the stratospheric circulation is better studied using the age of air as in Diallo et al. (2011). The 10-year average $CO_2$ profiles after removing the mean from the $CO_2$ are described in section 5.3. We have extended the top level to 42km in Fig. 6b.

7. ***A separate discussion Chapter at the end is necessary, especially an assessment and classification of the data set regarding the current knowledge. In particular to point out the "new" findings and a rating about the quality of the provided data set would be helpful. As announced in the Introduction (page 6, line 21) "The global (!) distributions of $CO_2$ and its transport in the UTS (!) are investigated . . .". Therefore, it would be nice to see some "global" charts, covering the entire stratosphere in this paper: The final sentence of your manuscript (page 21, lines 12-13) makes me***

*curious, but no respective illustrations are shown in this paper. Could you please show (and discuss!) some examples, not only to motivate using this comprehensive data set?*

The final section has been rewritten to emphasize the results. We do not fully understand the request of the reviewer as Fig.6 is an example of the global charts which are provided from our latitude x altitude product.

Minor points:

*Many acronyms are not explains, in particular in the Introduction.*

These acronyms are common. Avoiding them will remove the message that we want underline here and it keep the sentence shorter as well.

*Page 3, line 19: . . . increase global change . . .*

Done

*Page 4, line 14: . . .observations that have high resolution and are very localized, . . .*

Done

*Page 5, line 29 and in particular on Page 7, line 3: TRACZILLA is not explained (at least references are required).*

Done

*Page 10, lines 14-17: need a Revision*

Done

*Page 10, line 22: "It is assumed that the vertical mixing is fast . . ." Why? Reference is needed? How fast? Please explain.*

It's assumed the vertical transport is fast mostly driven by convection.

***Page 14, line 7: Units should be corrected.***

Done

***Page 15, lines 15-16: ". . . of four middle latitude stratospheric balloon flights . . .".***

Done

***Page 20, lines 4-5: "... based on 22 years of data ... over the last 11 years ...". Sentence should be revised.***

Done

***Page 21, line 3: "In the deep stratosphere we have found ... " no results are shown or discussed in detail.***

Done

***Figure 4: the chosen latitudinal bins do not exactly correspond to the coordinates of the stations providing the measurements.***

Done

***Figure 6: the box in the right figure (top right) says "2000-2011"; should be 2000-2010.***

Done

---

## Author Comment (AC2) · 29 Nov 2016

[acpd,hvmath]copernicus$_{discussionscolor}$

m.diallo@fz.juelich.de Answer to referee #1 Diallo, Legras, Ray, Engel and Anël

11

We thank referee #2 for his comments and suggestions. Comments by the referee are highlighted and followed by our answers.

1. ***The global product requires uncertainty estimates/bounds and a description of that derivation. This is essential.***

   We have estimated the uncertainties of the global CO2 distribution and added a subsection which describes the deviation (see Fig.6c ad sect. 5.3).

2. ***The manuscript needs to clarify the product for global CO2, that it is 2-dimensional (varies with latitude and altitude but not longitude). This was unclear earlier in the text, and only clarified in the final conclusion sentence. How this was generated in terms of where the receptors for the***

*trajectories were distributed in the stratosphere is unclear. How are the ini-
tial air parcels distributed in time and space, and are they run from different
longitudes and the results averaged in zonal means?*

The reconstructions is indeed produced by averaging over all parcels launched
on a latitude circle. Further grouping in latitude bins is made near the pole. This
is explained in section 3.1. and 3.2. The reconstructions of balloons and aircraft
measurements at given locations is based on multiple launching from the same
point and diffusive trajectories as explained in section 3.3.1

3. *Grammar and spelling should be checked throughout the manuscript. I
   noted many of the errors below, but it should be thoroughly proofread.
   Many sentences are awkward. The quality of the writing seems to dete-
   riorate even more in the final few pages, with basic typographical errors
   and usage of words that are not words.*

   This aspect has been hopefully improved.

*Specific comments:*

*Overall, the journal editor can comment on whether italics are or are not appro-
priate where used in this manuscript, generally when defining a term or acronym
(for example "tropical pipe" is italicized throughout the paper).*

The italics term have been removed from the text.

*L21: decreases with altitude... is nearly constant with altitude - contradictory.
Perhaps the authors intend that CO2 decreases with altitude from the UT to the
S but is nearly constant with altitude above 35 km?*

Yes that we meant here! The $CO_2$ concentration decrease up to 35km and above it is
nearly constant.

*Page 4: L 6: where recently assessed from balloon-based*
*L 28, qualitative? Perhaps better worded would be "shown qualitatively good agreement with in situ observations..." Last line and L1 of Page 5, awkward sentence*

We have rephrased this sentence

*Page 5:*
*L25 grammar*
*L18 this diffusivity effect*
*L29 models*
*I don't completely understand why Lagrangian models would not be subject to these problems if the underlying transport model is flawed.*

Grammar corrected
CTM are usually run at low resolution due to the numerical cost of chemistry. They are therefore highly diffusive and can generate spurious diffusive transport and mask barrier effects due to the dynamics. Lagrangian models are purely advective and not bounded by diffusion. Added diffusion is however necessary to represent missing turbulence in the advective winds but it is several orders of magnitude smaller than the spurious numerical diffusion of CTM.

*Page 6: L10: by scarcity are the authors referring to the scarcity of CO2 observations? confusing sentence, possibly because of wording /grammar errors.*
*L15: is the ERA-interim analysis used in any of the previously referred-to flawed CTM models that were unable to correctly capture transport?*
*L15 ERA-Interim definition should be moved up here from Line 20-21.*

This paragraph is rearranged and improved.

*Page 7: L2: the Lagrangian transport model TRACZILLA (Legras ..), a modified version....*
*L5: I see now that the Lagrangian model is calculating its own vertical motions, unlike perhaps the CTMs mentioned above, and that is why it performs better?*
*Line 15: Please clarify the assumption being made here: this corresponds to the assumption that the $CO_2$ in the troposphere (i.e. below this boundary condition) is constant?*

The vertical motion is derived from the archived heating rates of the reanalysis. This procedure is also quite common among the stratospheric CTM. The heating rates allow to better separate the horizontal (isentropic) motion and the vertical (cross isentropic) motion. They are also smoother, being accumulated quantities, than the vertical velocities which are instantaneous sampling (undersampling actually) of fast varying fluctuations. As 500 hPa is taken as the lower boundary of the domain, trajectories starting below (as it may happen only in high surface pressure regions) are discarded.

*Section 3.1, page 8. The use of clean-air data at the ground, no matter how clean the site, is a bit worrisome. Since Carbon tracker has also been used for the later years, how would it impact your results if you used a lower or higher CarbonTracker level? The 5km level is not only above the PBL, it is significantly higher than the other stations that you use in 1989-1999. It would be worthwhile to investigate the vertical gradient in CarbonTracker to see what kind of error you have just by choosing one level and assuming it is constant. Also it would be useful to look at CarbonTracker residuals against the NOAA North American aircraft network data at altitudes above the PBL, to see if it is a realistic depiction of $CO_2$ mole fractions in the free troposphere and upper troposhere (those profiles go to 8 km).*
*http://www.esrl.noaa.gov/gmd/ccgg/carbontracker/profiles.php*
*If residuals are small, then using the CT gradient might be a good way to determine the uncertainty on the assumption and the choice of only one level. CT can*

*also be used to investigate any longitudinal errors/differences.*
*It is not clear if the CarbonTracker mole fractions are considered as an average*
*over all longitudes, or for the specific grid cell where the back-trajectory initi-*
*ates.*
*L22-23: This is not a great description. CarbonTracker assimilates CO2 observa-*
*tions from atmospheric stations and optimizes underlying fluxes from the listed*
*sources (ocean fluxes, biosphere fluxes, fire and fossil fuel).*

The reconstruction procedure is well posed by using a given pressure surface as a
lower boundary. As far as vertical transport in the lower troposphere occurs in a matter
of days, this is not going to impact our results. Taking the CarbonTracker data at 5km
filters out the ground fluctuations and allows to use a fairly limited sample of trajectories
to provide a smooth reconstruction.

*Page 10: L19: I repeat my question from before, which could be addressed here*
*- it is assigned based on the lat/lon of the boundary crossing after 2000 but prior*
*to 2000 it is only the latitude that matters?*

Yes, the attribution of the CO2 concentration to the back-trajectories after 2000 is made
depending on the lat/lon position of the air parcel for a given time. We only averaged
the 3-hourly CO2 mole fraction from CarbonTracker in daily CO2 concentration that we
use to initiate the trajectories.

*Page 10-11: Are the particles in each bin spread evenly throughout? I.e. is their*
*initial location in the center of the bin or evenly distributed in time and space?*
*this may have been mentioned earlier but should be clarified here. (Page 11, line*
*6 discusses how many particles there are per bin, so this would be a good place*
*to discuss how they are distributed in time over the month and in space).*

The air parcels are distributed uniformly in longitude $2°/\cos(lat)$ and every $2°$ for the
latitude. At the end of each month, we distribute at each altitude level 10250 particles
which are driven with the ERA-Interim winds.

*Page 12, L10-15, this discussion of diffusion answers my previous question about the difference in the models - it could be briefly mentioned earlier in the paper to clarify this.*

Okay good! In this section, we calculate these reconstructions at a given location and therefore need to proceed differently than the global reconstructions.

*Page 13-14. Perhaps you could clarify why this 6-month dispersion of particles is required before the particles are tagged with the value of the global reconstruction. I would think you could just sample your global distribution at the location of the observations?*

As the local variability sampled by in situ measurements is mainly due to the chaotic properties of advection, it is necessary to integrate backward in time to reconstruct this variability from the grossly sampled global reconstruction. Launching only one parcel from each measurement point would reconstruct fluctuations growing with the backward integration time. Using a cloud of diffusive trajectories generates a reconstruction that does not depend on this integration time once it is long enough (see Legras et al., 2005).

*Page 14, Line 21, reference to figure needed in text here.*

Corrected

*Page 14, L23: This is the first reference to a CI for the reconstruction. How is it calculated/obtained? The CI or uncertainty is very important to anyone who would be using this product, and the methods for its calculation should be well documented in this paper. Presumably some of this comes from the particle/diffusion release of the 6-month trajectories from the flight, but there should be an uncertainty associated with the global CO2 product as well.*

The confidence interval of the reconstructions at a given location is clearly an upper bound of the uncertainty of the global product.

*Fig. 2: I would like to see differences in addition to the time series of mole fractions, or correlation plots with $R2$ statistics related to bias and/or RMSE; i.e. a more quantitative comparison. A plot or a table of statistics, or perhaps a single plot with all the flights colored differently, would not add too much length or bulk to the paper.*
*Fig. 3 shows the differences well enough that it seems fine, and statistics would be difficult to calculate for such a small sample.*

The correlation and $R^2$ statistics have been added in a separate figure.

*Page 15, L23: We have no information on the error calculation so this sentence is confusing.*

This paragraph has been rewritten.

*Page 16, L 13: CONTRAIL should be described somewhere (in the section where SOLVE and the balloon flights are discussed in Section 3.2).*

A paragraph describing CONTRAIL data have been added

*Page 16, section 5.3, why is CONTRAIL treated differently from Solve and the balloon observations, and no 6-month diffusion of particles is conducted? (this is the same as my earlier question about why that was done for those observations, there should be some explanation for this).*

The SOLVE and balloon observations are very localized compare to the CONTRAIL that has a large spatial and vertical coverage over several years as shown in Sawa et al., 2008. SOLVE and balloon observations were used to evaluate the ability to reproduce high resolution in situ measurements while CONTRAIL is a more direct test of the global distribution.

*Fig 4, some uncertainty bounds should be calculated and shown for the global initialisation using TRACZILLA. (could be based on differences from Contrail, or based on the spread of values of the trajectories that constructed each bin). Some error is likely also introduced by the choice of the level in CarbonTracker (looking at any vertical gradients in CT product could help quantify this, perhaps it is very small).*

We have calculated the standard deviation using the spread of values of the trajectories that constructed each bin.

*Page 16: Why was 15 days chosen, was it because that delay fit the data best?*

Yes! 15 days was chosen because that delay fit the data best at this height but we recovered the 2 months delay at 19-18km above the tropopause.

*Page 16, last line: give mean, standard deviation of differences?*
*Page 17, Line 6-7: in this period, discussed further in section xX.*
*Page 17, Line 18: awkward description of the biospheric CO2 seasonal cycle*
*Page 17, L19: CO2 concentration in the UT increases (subject /verb agreement in this sentence).*
*Fig 5, caption should indicate that the source of this data is the CO2 global re-construction*

Done

*Page 18:*
*L3: tropics*
*L13 maximum*
*L14 observations, also refer to CONTRAIL here to clarify this is the same data set used in the earlier comparisons*

*L25 isolates*
*L26 no comma after effect, also this is still awkward phrasing*
*L27 homogenizes*
*L28 containment*

corrected and sentences rephrased

*Page 19:*
*L9 profiles.... exhibit*
*L10 I don't think interspelled is a word, please rephrase (perhaps interspersed, or alternating?).*
*L16, processus (process)*
*L16, injects*

corrected and sentences rephrased

*Page 20*
*L7 that should be which*
*L11 "This good agreement demonstrates that the Lagrangian model (TRACZILLA) ..."*
*L23-25: awk sentences (high horizontal mixed and uniformise)*
*L17 extent*
*L29 troposphere*

corrected and sentences rephrased

*Page 21*
*L1 variability*
*L1 subject/verb (variability are)*
*L11 measurements should not be capitalized*

corrected and sentences rephrased

---

## Author Comment (AC3) · 29 Nov 2016

[acpd,hvmath]copernicus$_{discussionscolor}$

m.diallo@fz.juelich.de Answer to referee #1 Diallo, Legras, Ray, Engel and Anël

11
We thank referee #3 for his comments and suggestions. Comments by the referee are highlighted and followed by our answers.

*Major Comments:*

1. ***My only major point is the quality of the text (outline, titles of the sections and sub sections). I am not a native speaker but my feeling is that not only the structure of the paper but also the quality of many sentences can be significantly improved.***

    We have rephrase some of the titles and improve the discussions

**Minor Comments:**

1. **Abstract L 1**
   **please remove "relevant". You can only hope that this will be a relevant data set.**

   Done

2. **Abstract L 8**
   **I would replace "guided" by "driven"**

   Done

3. **Abstract L 11**
   **...with mid-latitude vertical profiles measured in situ from aircraft and balloons exhibit a remarkable agreement...(you should not give a complete description of the used data in the abstract)**

   Done

4. **Abstract L 17**
   **...out of the tropics to the mid and high latitude stratosphere (but mainly into the northern lowermost stratosphere around 15 km - is it not something that follows from your Fig. 6)**

   Yes! Rephrase

5. **Abstract L 21**
   **...and is nearly constant above 35 km (is it what you would like to say ?)**

   Yes exact! Rephrase

6. **P.4 L3**
   **These studies.... - please rewrite this sentence**

   Rephrase

7. ***P.4 L 12***

   ***...to help to validate the stratospheric representation in global CTMs...***

   Rephrase

8. ***P.4 L14***

   ***...to the very localized in situ observations which have high spatial resolution, a large spatial...***

   Rephrase

9. ***P.4 L18***

   ***Chadin et al showed.... (please rewrite this sentence)***

   Rephrase

10. ***P.5 L5***

    ***...are also weak... (what do you mean ?)***

    Rephrase

11. ***P.5 L21/22***

    ***"model's lack of realistic stratospheric influence" - not clear, please explain***

    Rephrase

12. ***P.6 L10***

    ***The small scale variability....and the scarcity of suitable observations ...(I would recommend to reformulate this sentence)***

    Rephrase

13. ***P.7 L14 Trajectory starting....(you are using backward trajectories, so maybe you would like to write: "Trajectories reaching the boundary layer during the backward integration..."***

The reconstruct with model is done in two steps:

1-) We calculate the trajectories by integrating them backward in time.

2-) We post-treat these trajectories by assigning them within CarbonTracker $CO_2$ in the troposphere at 500hPa.

14. ***Section: Data***

    ***You should shortly describe here the aim of both upcoming subsections***

    Done

15. ***Section 4 and 4.1***

    ***I think, you should use a different title like "Reconstruction of CO2" and describe it accordingly. "Initialization" is a very misleading term. So you use backward trajectories plus Carbon Tracker/WDCGG data in the boundary layer to reconstruct CO2 everywhere in the UT/S region. Please reformulate the text between L10 and L20....***

    Title and sentences rephrased

16. ***P.11 L14***

    ***Maybe you should change b0 (which is too close to b) to something different.***

    Done

17. ***P.12 eq (2) and (3)***

    ***Maybe you should avoid to introduce Corr, i.e. use only one formula in two lines***

    We have used one line formula as suggested.

18. ***P.11 eg. (1)***

    ***You also did not clearly explain that you need your eq. (1)-(3) only for CO2 reconstruction cases which are older than 10 years and you do not have any***

*information from the backward trajectories. Maybe you should reformulate some sentences...*

Yes right. We include these comments explicitly.

19. **P.12 4.2.1**
    **Once again: for me it not a "flight track initialization" but much more a "Lagrangian reconstruction of CO2 along the flight track" - maybe you should reformulate it**

    Yes! Rephrase

20. **P.12-13**
    **You repeat here many arguments and formulations from Legras et al., 2005. I do not think this is necessary. I strongly recommend to remove this part and cite the original papers. Instead of this, you should better explain your eq (7), i.e. how displacement in the geometric space is related to a displacement in $\theta$–space.**

    We have shortened this discussion.

21. **P.14, 4.2.2**
    **Once again, please use the term "Reconstruction" in the title of this section and change accordingly the following text...**

    Done!

22. **P.14 5.1**
    **Here, I miss any reference to Fig 2.**

    Corrected

23. **P.14 L 24**
    **Your abbreviations of the dates are not clear and maybe you should the notation like 26th February 2000, etc.**

Interactive
comment

Figure notation is included in order to overcome this.

24. ***P.15 L 6***
***aging vortex core***

    Corrected

25. ***P.15 L 7***
    ***"corrective step in this instance" - I do not understand what you mean***

    This was belonging to the applied correction to the remaining air parcels into the stratosphere after the 10-yr backward integration using equations 1-3.

26. ***P.16 L 6***
    ***"accurate" - please remove it***

    Done

27. ***P.16 L 8***
    ***...from the global reconstruction calculated by...***

    Rephrase

28. ***In Fig 4a and 4b you denote the regions as tropospheric and stratospheric boundaries. However, in the model there is only one lower boundary prescribed by the Carbon Tracker values. You should exactly say what you mean with your boundaries. For me, 4a validate your model in the middle troposphere around 7-9 km and 4b in the region between 16-17 km. Please clarify this point.***

    What we mean here by tropospheric boundaries is that we evaluate our model in the middle troposphere around 7-9 km which is not far away our trajectories $CO_2$ assignation and by stratospheric boundaries the near tropical tropopause layer where the air enters the stratosphere. We have improved it.
29. *P.17 L 3*
   *with respect to*

   Done.

30. *P.17 L 4-7*
   *please give a more detailed explanation of the discrepancies. "as it will be confirmed shortly" could be replaced by "We discuss this point in the next section".*

   Need to be done

31. *P.17 L 12*
   *...derived from our Lagrangian reconstruction*

   Done

32. *P.17 L 19*
   *grows*

   Done

33. *P.17 L 21*
   *in the southern hemisphere*

   Done

34. *P.18 L 4*
   *propagates*

   Done

35. *P.18 L 5*
   *is removed from the atmosphere due to...*

   Done

36. ***P.18 L 6***
***into the lower stratosphere driven by the lower branch...***

    Done

37. ***P.18 L 9***
***west side (can you give more detail how the Asian monsoon anticyclone contributes to this transport***

    We notice just the Asian monsoon anticyclone contributes to this transport as well discussed in Park et al., 2009, Randel et al. 2010 and Pan et al., 2016 which were specifically focused on this contribution on pollutant transport. In our fig.6(a,b), we can clear see the "cheminay" further during the summer months June and July between 10-40 N.

38. ***P.18 L 13***
***which is maximum...***

    Done

39. ***P.19 L 1-2 localized gradient, etc. - please give a more detailed explanation***

    Done

---

## Author Comment (AC4) · 29 Nov 2016

[acpd,hvmath]copernicus$_{discussionscolor}$

m.diallo@fz.juelich.de Answer to referee #1 Diallo, Legras, Ray, Engel and Anël

11

[Figure]
We thank referee #4 for his comments and suggestions. Comments by the referee are highlighted and followed by our answers.

*Major Comments:*

1. ***The evaluation of the data set is very specific and done on the basis of very limited and selected case studies: SOLVE only covers the high Arctic in winter 1999/2000. Similarly the four profiles are arbitrary snap shots, showing disagreement in one case, which is not even tried to be explained sufficiently (p.15).***

   The SOLVE dataset is not limited to the polar regions as we have included test

flights and transit flights in the extratropics and the mid-latitude, that is six flights out of 12. Moreover, testing CO2 reconstructions in the polar region during winter is, a priori, the most difficult situation since the polar air is old and should cumulate all the errors in the transport. Regarding balloons, the number of flights with high quality CO2 data is unfortunately very small and only four are available in the 2000-2010 period. We have also extensively used CONTRAIL data which are partly stratospheric at mid and high latitude over a range of 4 years. Therefore we did our best to evaluate our dataset with the available observations. The sparseness of CO2 in the stratosphere is one of the reason of our work. See also our answer to item 3 below regarding the comparison with estimates of the age of air.

2. ***Further the discussion of the data and differences to literature is very superficial: The time delay between the cycles is mentioned but not appropriately discussed, since Boering et al., 1996 use N2O = 310 ppbv as reference values to determine CO2 at the tropopause, whereas in the manuscript a fixed altitude level is used, which is at or even below the tropical tropopause. This is not considered in the manuscript. Instead the authors conclude "...We recover a two-month delay at higher altitude in the layer 18–19km (not shown). The origin of this discrepancy is unclear but is perhaps due to the fact that previous studies merge measurements in the deep tropics and the subtropics."***

There is actually no contradiction with Boering et al., 1996, as our 15 days delay is valid in a layer under the tropopause where the air is renewed by convection while the estimate of Boering et al. is associated with the lower tropical stratosphere where the age of air with respect to the crossing of the tropopause grows rapidly with altitude. It is about 6 months at 20 km (see Diallo et al., 2012, where the prediction of our trajectories is shown to fit the ER-2 measurements) and it is therefore not surprising that we find a delay of a 2 months in the 18–19 km layer

in agreement with Boering et al. We have modified the discussion.

3. ***This is not satisfying for a quantitative reference data set over 10 years. Given the statements in the abstract (l.3-6: "...This product can be used for model and satellite validation in the UT/S, as a prior for inversion modelling and mainly to analyse a plausible feature of the stratospheric-tropospheric exchange as well as the stratospheric circulation and its variability..."), a quantitative assessment and evaluation also of the stratospheric data is required, e.g. by using age of air diagnostics. This would allow for comparison with other data sources or diagnostics (e.g Eyring et al., 2006; Haenel et al., 2015). For this the authors could also include SF6 to their analysis, since the authors conclude that their results hold for any long-lived species (p.20, l.14). Even if SF6 cannot be directly included, the age of air information can be inferred from the data. This would provide a quantitative comparison to evaluate the results on the basis of CO2 and the consistency of the results within the model. It would further help to evaluate their stratospheric data using satellite observations of e.g. MIPAS SF6 in regions where the in-situ data are sparse or absent. Even without SF6 the calculation of age of air allows for comparison with other data sets.***

The same trajectories have been used in Diallo et al. (2012) to calculate the mean age of stratospheric air which has been compared with age estimates based on CO2 but also SF6 and N2O measurements, and also with the GEOSCCM model. It should be observed that SF6 is photolysed at high altitude and therefore cannot be as easily interpreted as CO2, especially at high latitude during winter where a large amount of air has descended from high altitude [more detail see Stiller et al., 2008, 2012].

***Minor Comments:***

1. ***Abstract: Last sentence: Please clarify the sentence and specify: there's a contradicition: decrease or constant? Decrease of CO2 to 35 km or constant, constant with altitude above?***

   Done

2. ***Introduction: Do you need the first sentence?***

   Removed

3. ***p.3, l.2-4: The increase of green house gases does not increase tropical upwelling mass flux, it is the effect on atmospheric temperature structure and wave propagation.***

   Right. Corrected.

4. ***p.3, l.6: stratosphere instead of atmosphere? CO2 is destroyed in the upper atmosphere.***

   Done

5. ***p.4, l.7: Here you need to mention the Engel et al., (2009) study - not on the previous page (l.25), since it is not beased on airborne measurements.***

   Done

6. ***p.7, l.2: You forgot 'TRACZILLA'***

   Done

7. ***p.8, l.20-22 (and l.10 ff): What does this mean: Similarly to 1989-1999.... only at 5 km ? Please specify the altitude criterion of the selected stations for 1989-1999. How many stations contribute? It would be good to show a 3D distribution of the boundary condition: e.g. a zonal mean plot with latitude as y axis, time as x-axis and CO2 as iso surface above to see the global distribution and allow for comparison with e.g. the NOAA CCGG data.***

Corrected. The 1989-1999 is based on ground stations that are far from sources. The criterion to select the ground stations is that the elevation is high enough to neglect the variability due to localized sources at ground level.The ground boundary condition would project on a 2D map but it contains a large amount of variability which is not relevant here. We choose the 500 hPa surface as a boundary condition after 2000 in order to filter out the surface fluctuation and reduce the number of needed trajectories. The CarbonTracker CO2 which is used to initiate our trajectories is perhaps the best currently available tropospheric CO2 in the range 2000–2014 because it assimilates all available observations, including CCGG data, to produce a 3D distribution of CO2 in the troposphere. Therefore a comparison with CCGG data will be redundant and anyway we do not aim at outperforming CarbonTracker in its domain of validity. Our dataset applies above 500 hPa only. The NOAA CCGG webpage higlights the Carbontracker CO2 http://www.esrl.noaa.gov/gmd/ccgg/.

8. ***p.10, l.11: Wrong sentence? Something is missing...?***

   corrected

9. ***p.11, l.3: Is a zonal mean calculated? What are typical numbers of parcels per box?***

   We launch 10.255 particles per levels and we have 30 levels. The discretization is described in section 2. longitude=2/cos(lat) and latitude=2 degrees. In the tropics we have 180 particles that decrease as 2/cos(lat).

10. ***p.12, l.4: Chapter title: The term 'validation' is used, but one can't validate the results, since you can have agreement for the wrong reasons. Therefore I suggest the term 'evaluation'.***

    Okay... Even if here it's not the case.

11. ***p.12, l.15 ff.: The exponential factor b clearly depends on the driving data set. Does the exponential factor b further depend on the choice of the trajectory model and needs in principle to be determined for each individual trajectory model?***

    The exponential factor is a statistical quantity which can only be defined for a large ensemble of trajectories and not for a single one. It is defined for the whole stratosphere at a given date but varies vey little. We made trials of defining by averaging in time for a given set of latitudes and altitudes with negligible effects.

12. ***p.13, eqn.4/5: Please clarify the notation of vectors, scalar products and sclara quantities. Why is the 't' in bold font?***

    Corrected.

13. ***p.13, l.11: ".. kappa defined by the user..." Is k (kappa) chosen to have the same value in the whole atmosphere? Please add a word, which values have been selected or how a user has to define Kappa.***

    Yes the diffusion is chosen to be the same for the whole atmosphere. We have replaced this discussion by a reference to Legras et al., 2005 where a thorough discussion of the determination of diffusion is provided. It should be recalled that diffusive dispersion of trajectories is only effective during the first 3 or 4 days of the backward integration, hence the chosen diffusivity has to be valid in the lower stratosphere only.

14. ***p.15, l.19-23: Please explain, how a cold front (which is a tropospheric feature) can affect the CO2 a 25 km altitude. The inset in Fig.3. is too small and does not contain a legend. It is further unclear, why the PV gradient should be associated with a CO2 gradient. This paragraph sounds very weird or almost wrong.***

There is nothing weird here but the mere usage of common concepts in dynamic meteorology. When submited to chaotic stirring, all long-lived tracers that are bound to preserve their tracer-tracer relation tend to align their contours and therefore the $CO_2$ contours are likely to follow the PV contours and to exhibit high gradients at the same location of high PV gradients (as would ozone or $N_2O$ do as well). A cold front is the surface signature of a deep structure that penetrates the stratosphere. The map shows the PV distribution at 18.5 km well above the mid-latitude tropopause located at about 11 km. A similar case has been studied in depth by Pisso and Legras, 2008. See also Miyazaki et al., 2009.

15. *p.15, l.27-29: "... The mean in situ $CO_2$ from observations is much more spread in the high latitude profile (44 N) above 15 km. There is not a clear explanation about these observed fluctuations on the in situ $CO_2$-profile.": What do the authors want to say with such a statement? What does this mean for the comparison? What does it mean for a data set, which is intended to serve as a reference for model evaluation from 2000-2010, if one out of four stratospheric profiles does not fit the observations?*

It is well known that modelling tracer distribution is highly prone to transport errors in the region of large gradients. It was shown by Pisso and Legras, 2008, that the global pattern is weakly affected but a small displacement can in this case generate important deviations. Therefore it is expected that the test will fail in such a case which cannot be used as a reference. The discussion has been rewritten.

16. *p.19, l.20 and l.23: What is meant with "the subtropical barrier" in this paper? Do you mean the subtropical jet at the tropopause, which exhibits a seasonality with weaker PV gradients and high permeability in summer? The subtropical barrier normally denotes the boundary of the (leaky) tropical pipe in the overworld (e.g. Palazzi et al., 2009), which does not show the same variability as the STJ and has a different generating mechanism.*

By subtropical barrier we mean "the subtropical jet at the tropopause, which exhibits a seasonality with weaker PV gradients and high permeability in summer". This is a common terminology in dynamic meteorology where it has been known for a long time that the jet centered at 200 hPa and about 30N and S inhibits exchanges between the upper tropical troposphere and the lowermost extra-tropical stratosphere.

17. ***Fig.2: Please include potential temperature along the flight track as additional information. Otherwise the information on the plots is without any relevance for a scientific interpretation and an estimate of the quality of the model capabilities. Does the gray area refer to the variability of the data in a bin? If not, how is the error calculated?***

    Potential temperature along the flight-track has been included. The gray area is 95% confidence interval estimated from the model results.

18. ***Fig.5: The continuous color bar is not consistent with the figures, which have discrete colours. Please provide a discrete color bar legend.***

    Done.

---

## Referee Report (RR1)

Review of the paper (revised version):

"Global distribution of CO2 in the upper Troposphere and Stratosphere"

written by M. Diallo et al.

**General:**

The paper in its revised version improved a lot. The most important results (3D Lagrangian reconstruction of CO2, its validation, analysis of the propagation of the annual cycle into the stratosphere, formation of inverse gradients of CO2) are well documented. The paper is supported by very good figures. However, still some "polishing work" is necessary. Especially the introduction is too long and not really focused on the following text. The paper may be acceptable after some "editorial and polishing work" by including a native speaker. One major point and some minor points are listed below.

**Major points:**

1. Your introduction is too long. E.g. the detailed description of all in situ campaigns/satellite observations and their results (which are not used in the paper) is not necessary, e.g. P3/L27 - P4/L28 - here is some potential to shorten your introduction.

**Minor points:**

1. P2 /L1
   ...a new 3D data database...of carbon dioxide (CO2) extending from...

2. P3/L3-6
   satellite validation, inverse modeling - you are almost not talking in your paper about these points. However, you talking much more about the circulation, inverse gradients, seasonal cycle. Please reformulate this sentence.

3. P3/L2
   ...can also be diagnosed...

4. P3/L12
   ...anthropogenic emissions, deforestation, biomass and fossil fuel burning...

5. P3/L14
   Tans and Keeling, 2015 - I did not find this citation in your literature list

6. P3/L15
   "that represent" x2

7. P3Ł20
   ...were hold...

8. P3Ł28
   The SPURT campaigns were...

9. P6Ł21
   ...section section...

10. P7Ł4-5

    ...to assign CO2 to air parcels transported along the backward trajectories. During the 1989-1999 time period, data from...are applied. The WDCGG is an international...

11. P7Ł14-20

    Remove some repetitions. This part of the text should be reformulate

12. P8/L6

    "CO2" is not correctly written

13. P8/L19

    ...in the northern polar regions

14. P9/L17

    ...in the Northern Hemisphere...

15. P9/L19

    "air particles" or "air parcels" (please unify this notation everywhere, I would prefer "air parcels")

16. P9/L20

    ...and integrated backward in time

17. P10/L7

    ...the whole stratosphere at any latitude and longitude, 30 levels...

18. P10/11-14

    "Trajectory starting" - you consider backward trajectories. Please reformulate this part with too much technical jargon.

19. P10/L19

    "is impacted" - it does not sound good. Maybe "was hit during the backward integration"

20. P11/L8

    ...and that remain within... - and staying within

21. P11/L11-12

    I do not understand your sentence with standard deviation. Maybe 2 sentences would be better

22. P12/3.3

    Instead of "evaluation" I would recommend to use "validation" in the title and everywhere in the text

23. P12/L22

    ...that a single trajectory processed by... - sounds very strange. Maybe: ...that a single probe can be understood as a mixture of sub-parcels...

24. P13/Formula (3)

    You should also explain u(X,t)

25. P13
   The reconstruction of CO2 observations is with mixing and the global reconstruction is without mixing. I think you should explain it little bit.

26. P14/L6 and L10
   same sentences

27. P14/L19
   ...the two curves almost superimpose but for small-scale fluctuations... - you probably mean: ...the two curves agree fairly well even for small-scale fluctuations...

28. General
   For the sub-panels of the figures you sometimes use (a) or a. Please unify your notation

29. P16/L16
   ...and we recover a delay of 2 months... - ...and we diagnose a delay of 2 month...

30. P17/5.1
   ...and lowermost stratosphere (please unify large and small characters in titles)

31. P20/L2 and 4
   "inherited" - please replace by a different verb

32. P21/L4
   "to uniformise" - to be uniformly mixed at isentropic surfaces

33. P21/L9
   "in opposition" - opposite

34. Fig 1
   The upper green line is shifted

35. Fig 4
   Black curves, Green squares without brackets. You write 17 September but in your figure I see 1th September, please correct

36. Fig 5/L5
   average of what

---

## Author Response (AR2)

**Diallo, Legras, Ray, Engel and Anẽl**

**Correspondence to: m.diallo@fz.juelich.de**

Dear Editor, Dr. Gabriele Stiller

We have made modified the manuscript in order to seriously consider the reviewers suggestions and comments. These changes include: a new figure 4.b concerning the discrepancies between the reconstructed CO$_2$ profile in 9 October 2001 and the balloon observations. The language of the manuscript still has been improved by native speaker. The abstract has been entirely rewritten. The initial mean error from CarbonTracker has been added in the text.

Best regards,
Mohamadou Diallo (on behalf of the co-authors)

**Answer to anonymous referee #1**

We thank referee #1 for his comments and suggestions. Comments by the referee are highlighted and followed by our answers.

**Minor points:**

1. *The paper has improved a lot. This paper is ready for publication now. There is only one (still open) point which I have raised in my first report:*
   *page 16, line 7 (and following): Why do you use a asymmetric (with respect to the equator) latitudinal band (10S-20N)? And it does not fit with American Samoa (14S)!?*

   We wanted to have the same latitude bin 10S-20N for both timeseries (Fig.5 (a,b)) because the CONTRAIL data have less observations in latitude bin 10S-20S and extending to 4-10 degrees in the south does not change the inter-comparison in Fig.5b at least at that level 16-17km.

**Another minor point:**

*There are some typos which I found during reading the manuscript*
  The manuscript was corrected by a native speaker and also sent for English correction service.

  *page 5, lines 14 and 20: brackets with respect to Shia et al.*
  Done
  *page 8, line 6: CO-2*
  Done
  *page 8, line 20: vortex .*
  Done
  *page 9, line 23: TRACZILLA. (Legras...*
  Done
  *page 15, line 24: Fig.4 also*

Done

**Answer to anonymous referee #2**

We thank referee #2 for his comments and suggestions. Comments by the referee are high-lighted and followed by our answers.

1. *Awkward sentence structure still needs to be addressed.*

   The manuscript was corrected by a native speaker and also sent for English correction service.

*Specific comments:*
*Abstract: some awkward wording in here*
The abstract has been rewritten.
*p 1 L14: more readily than what*
Corrected
*p3:*
*L16 represents is redundant*
*L18-20 grammar (Despite its potential ... our knowledge) (should modify Co2, not knowledge)*
Corrected
*p5 L7 - specify "major disagreements between the model and observations occur in winter"*
Corrected
*p6 L20 section*
Corrected
*p7 how many and which sites? aircraft obs used at all? specify quantitative criteria for choosing sites ("high enough elevation to neglect variability..." is not quantitative). This was raised by reviewer 4 and was not addressed in the first round. The boundary condition could be shown or over time at given latitude, showing the condition as it goes from 1999 to 2000.*

As mentioned in the text page 7, we used only ground stations which are selected depending on their location (height, near city, Island, Ocean). There are 61 sites distributed in 30 latitude

bins: 6 stations in 60-90S, 7 stations in 30-60S, 8 stations in 0-30S, 10 stations in 0-30N, 23 stations in 30-60N and 7 stations in 60-90N. The selection criteria is not quantitative because it depends on which station is considered. In the main land close to the big cities like in China we only considered stations (Lulin: 2867m, Mt. Waliguan: 3810) with altitude above 2867m. For some stations located on Island and near the coast (Guam: 2m, Mauna Loa: 3397m, Black Sea: 3m, ...), the criteria based on altitude of the station is not necessary because these stations are far from strong local sources.

*p7 L21 Is there a discontinuity in time from switching between the two initialization products?*

There is not a discontinuity in time from switching between the initialization products because our trajectories integrate the contribution of sources over time.

*p8 L5-9. This is a good explanation of what effect the pre-2000 initialization might have. Although this section notes that the uncertainty prior to 2005 should then be higher, later on in the paper (p19) this is not mentioned - this should be incorporated somehow in the uncertainty analysis. I would recommend an uncertainty be placed on the initialization values that is based on uncertainties in carbon tracker and in the confidence of the pre-2000 initialization method. Then it could be added (in quadrature) to the uncertainty from the spread of the trajectories.*

The initial errors from the CarbonTracker (CT) are not included in the standard error estimated here. We consider CT as truth and its initial error will constitute additional error bar for final product. As demanded, here we estimate the monthly uncertainty (Fig. 6c) induced by the lagrangian model calculations. As shown, the model errors are quite small then any additional errors will be the CT errors during the initialisation. The mean error of CT on initial value is less tha 1.25 ppmv and should be added in quadrature to the standard error of the mean.

*p13 L16: the mean value over what location? I was under the impression that a single particle was released (for the ER-2 flights) at every one location (one particle per location)? Or are there multiple particles launched per flight measurement?*

*L17 how is the Confidence interval estimated? Presumably from the standard deviation of*

*the particle sampling over some number of particles - how many per location?*

The zonal mean value of CO2. Not a single particle was released but ensemble of particles: 200 for ER-2 at every one location. The confidence interval was estimated from the dispersion of the ensemble of air particles.

*Fig3. This is a big improvement over the previous version of the comparison. Note in caption that the dashed line is the 1:1 line. I don't see why both the correlation coefficient and $R^2$ are needed if one is just the square of the other?*

Yes. It's repeating. We have remove the correlation coefficient and keep the $R^2$.

*Fig 4. What does EI mean in the legend? Perhaps it should be CO2-model or reconstruction. In the caption as well, I would describe the black line as the vertical profiles from the model-reconstructed CO2.*

EI mean ERA-Interim. We modified the legend and corrected the caption.

*p14 and figures 3& 4: how is the CI calculated? Is only the scatter of particles considered, or is there a different error analysis being done here to account for additional possible sources of uncertainty?*

The confidence interval was estimated from the dispersion of the ensemble of air particles.

*p14-15 units should be on Delta throughout this section - this must be in ppm. Also in caption of Fig 3.*

Corrected in the caption and text.

*p19: uncertainty. line 5: why average over 11 years? Shouldn't each point in the global distribution, which is monthly, have its own uncertainty? I see that this average is used in the figure, but for the product, the uncertainty should exist for every month for every year, yes? Also clarify if this is a standard error or standard deviation (i.e. did you divide by the square root of the number of trajectories or not?). I believe this should be the standard error. See comment above about incorporating a second error by assigning an uncertainty on the initialization values as well.*

Yes it's the standard error of the mean. We averaged the uncertainty in the paper to have only one additional panel instead of 11x12 panels. Of course, the uncertainty exists for every month for every year. Yes, it is the standard error of the mean.

*Do the longer transit times have higher standard deviations because the trajectories have more time to spread out more essentially, so they hit the 500 hPa boundary at different latitudes and times, giving larger spread? I might add a sentence to explain this.*

Larger mean transit times (mean age) indeed mean also a larger variance of the transit and spread of the origin.

*fig 7a caption: black triangles, not orange. Approximately how many CONTRAIL profiles were averaged here? (over the whole month and latitude band?).*

The CONTRAIL data are given along the flight track no as flight. For 2007 there are 733 305 points over whole flight track. The profile construction is based on sampling for each given month, altitudes and latitude (here, May and Aug., 50-60N, 5-13km) all points available and averaged it. As the plane flies longer in cruiser it's natural to find more point in the altitude 10-13km than 4-5km for examples. Just to give an idea the sampling variations for May at 4-5km we have 427 points and at 11-12km we have 3638 points. The altitude sampling is 1km.

*p21 L4: uniformise*

Rephrased

**Answer to anonymous referee #3**

We thank referee #3 for his comments and suggestions. Comments by the referee are highlighted and followed by our answers.

*Major Comments:*

1. *Your introduction is too long. E.g. the detailed description of all in situ campaigns/satellite observations and their results (which are not used in the paper) is not necessary, e.g. P3/L27 - P4/L28 - here is some potential to shorten your introduction.*

   These sentences have been removed from the manuscript.

*Minor Comments:*

1. *P2 /L1*

   *...a new 3D data database...of carbon dioxide (CO2) extending from...*

   Done

2. *P3/L3-6*

   *satellite validation, inverse modeling - you are almost not talking in your paper about these points. However, you talking much more about the circulation, inverse gradients, seasonal cycle. Please reformulate this sentence.*

   Done

3. *P3/L2*

   *...can also be diagnosed...*

   Done

4. *P3/L12*

   *...anthropogenic emissions, deforestation, biomass and fossil fuel burning...*

   Rephrased

5. *P3/L14*

   *Tans and Keeling, 2015 - I did not find this citation in your literature list*

   Corrected

6. *P3/L15*

   *that represent x2*

   Rephrase

7. *P320*

   *...were hold...*

   Rephrase

8. *P328*
   *The SPURT campaigns were...*

   Rephrased

9. *P621*
   *...section section...*

   Corrected

10. *P74-5*
    *...to assign CO2 to air parcels transported along the backward trajectories. During the 1989-1999 time period, data from...are applied. The WDCGG is an international...*

    Rephrased

11. *P714-20*
    *Remove some repetitions. This part of the text should be reformulate*

    Done

12. *P8/L6*
    *CO2 is not correctly written*

    Done

13. *P8/L19*
    *...in the northern polar regions*

    Done

14. *P8/L19*
    *...in the northern polar regions*

    Done

15. *P9/L17*

    *...in the Northern Hemisphere...*

    Done

16. *P9/L19*

    *air particles or air parcels (please unify this notation everywhere, I would prefer air parcels)*

    We have replaced by air parcels everywhere.

17. *P9/L20*

    *...and integrated backward in time*

    Corrected

18. *P10/L7*

    *...the whole stratosphere at any latitude and longitude, 30 levels...*

    Rephrase.

19. *P10/11-14*

    *Trajectory starting - you consider backward trajectories. Please reformulate this part with too much technical jargon.*

    First we integrate backward the air parcels. After the backward integration, the trajectories are initialized depending on their position compared to the boundary condition. That is why we use the term "Trajectory starting" instead of "Trajectory ending".

20. *P10/L19*

    *is impacted - it does not sound good. Maybe was hit during the backward integration*

    Corrected

21. *P11/L8*

    *...and that remain within... - and staying within*

    Corrected

22. *P11/L11-12*
    *I do not understand your sentence with standard deviation. Maybe 2 sentences would be better.*

    Rephrased

23. *P12/3.3*
    *Instead of evaluation I would recommend to use validation in the title and everywhere in the text*

    The reviewer-4 suggestion in the previous reviewing: "Chapter title: The term "validation" is used, but one can not validate the results, since agreement can be reached for the wrong reasons. Therefore I suggest the term "evaluation" ".
    We have changed again "evaluation" to the initial wording "validation.

24. *P12/L22*
    *...that a single trajectory processed by... - sounds very strange. Maybe: ...that a single probe can be understood as a mixture of sub-parcels...*

    There is same mixing as we average in longitude the contribution of many air parcels.

25. *P13/Formula (3)*
    *You should also explain u(X,t)*

    Done

26. *P13*
    *The reconstruction of CO2 observations is with mixing and the global reconstruction is without mixing. I think you should explain it little bit.*

    Done

27. *P14/L6 and L10*
    *same sentences*

    Done

28. *P14/L19*

    *...the two curves almost superimpose but for small-scale fluctuations... - you probably mean: ...the two curves agree fairly well even for small-scale fluctuations...*

    Rephrase

29. *General For the sub-panels of the figures you sometimes use (a) or a. Please unify your notation*

    Corrected

30. *P16/L16*

    *...and we recover a delay of 2 months... - ...and we diagnose a delay of 2 month...*

    Done

31. *P17/5.1*

    *...and lowermost stratosphere (please unify large and small characters in titles)*

    Done

32. *P20/L2 and 4*

    *inherited - please replace by a different verb*

    Rephrased

33. *P21/L4*

    *to uniformise - to be uniformly mixed at isentropic surfaces*

    Done

34. *P21/L9*

    *in opposition - opposite*

    Done

35. ***Fig 1***
    ***The upper green line is shifted***

    Figure corrected.

36. ***Fig 4***
    ***Black curves, Green squares without brackets. You write 17 September but in your figure I see 1th September, please correct***

    Corrected

37. ***Fig 5/L5***
    ***average of what***

    Done

**Answer to anonymous referee #4**

We thank referee #4 for his comments and suggestions. Comments by the referee are highlighted and followed by our answers.

*Minor Comments:*

1. ***The only remaining (minor) point is the discussion of the balloon profile in the vicinity of an upper level trough (which indeed might be linked to a surface cold front), which shows a deviation between reconstruction and measurements. In the manuscript, the whole stratospheric column up to 35 km shows differences between observations and reconstruction. If a mismatch of the exact location of simulation versus observation is the reason for this (as in Pisso et al., 2008), this should be evident in vertical cross sections of $CO_2$ (or other quantities). I still doubt that a PV gradient at 70 hPa affects the stratosphere up to 35 km. Is it possible, that the data have a problem? The case in Pisso et al., 2008 is different since it shows, that a layered (eventually filamentary structure) of approx 1-2 km vertical extent is not correctly captured.***

In the absence of error bar on this balloon profile, it is not straight forward to state what is creating this spread on the Fig.4b. To do so, we have created a square box around this profile and reconstruct 8 vertical profiles which surrounded the initial profile. We have not found any big scatter among these reconstructed profiles which can explain the spread. In addition, for this balloon profile we have reconstructed the mean age and compared with SF6 and CO2 mean age derived from this balloon observation. The mean age derived from CO2 measurement show same dispersion as in CO2 profile but the SF6 mean age does not show such scatter. This suggests that this CO2 profile is questionable.

2. *In general (as I stated in my first review), a data set like the one which is presented here, is of great importance and highly appreciated. Therefore I see a strong need to correct for an appropriate use of the English language.*

The manuscript was corrected by a native speaker and also sent for English correction service.

---

## Author Response (AR3)

**Diallo, Legras, Ray, Engel and Anẽl**

**Correspondence to: m.diallo@fz.juelich.de**

**Dear Editor, Dr. Gabriele Stiller**
**In order to seriously consider your suggestions we have removed this statement from the abstract, increased the legends in Fig. and 5 as well as corrected the typos in these pages: page 15, line 9, page 21, lines 4/5 and caption Fig.6.**

**Kind regards,**
**Mohamadou Diallo (on behalf of the co-authors)**